# Notch-dependent and -independent transcription are modulated by tissue movements at gastrulation

Julia Falo-Sanjuan, Sarah Bray*

Department of Physiology, Development and Neuroscience, University of Cambridge, Cambridge, United Kingdom

**Abstract** Cells sense and integrate external information from diverse sources that include mechanical cues. Shaping of tissues during development may thus require coordination between mechanical forces from morphogenesis and cell-cell signalling to confer appropriate changes in gene expression. By live-imaging Notch-induced transcription in real time, we have discovered that morphogenetic movements during *Drosophila* gastrulation bring about an increase in activity-levels of a Notch-responsive enhancer. Mutations that disrupt the timing of gastrulation resulted in concomitant delays in transcription up-regulation that correlated with the start of mesoderm invagination. As a similar gastrulation-induced effect was detected when transcription was elicited by the intracellular domain NICD, it cannot be attributed to forces exerted on Notch receptor activation. A Notch-independent *vnd* enhancer also exhibited a modest gastrulation-induced activity increase in the same stripe of cells. Together, these observations argue that gastrulation-associated forces act on the nucleus to modulate transcription levels. This regulation was uncoupled when the complex linking the nucleoskeleton and cytoskeleton (LINC) was disrupted, indicating a likely conduit. We propose that the coupling between tissue-level mechanics, arising from gastrulation, and enhancer activity represents a general mechanism for ensuring correct tissue specification during development and that Notch-dependent enhancers are highly sensitive to this regulation.

*For correspondence:
sjb32@cam.ac.uk

**Competing interest:** The authors declare that no competing interests exist.

## Editor's evaluation

In this manuscript, Falo-Sanjuan and Bray provide an elegant set of experiments investigating how cell movements modulate Notch transcriptional activity during *Drosophila* gastrulation. Through a detailed and convincing analysis of live transcriptional reporters and gastrulation-defective mutants, they report a clear example of tissue movements affecting enhancer activity. The idea that morphogenetic movements can regulate the genome through mechanical changes in the nucleus is intriguing and important and is of broad interest to cell and developmental biologists.

## Introduction

Cells continuously sense and respond to their environment. This occurs via signaling pathways that detect and respond to external stimuli, such as morphogen gradients or direct cell-surface ligands. During development, information routed through these pathways feeds into networks of transcription factors that coordinate cell fates to pattern and regulate differentiation. At the same time, cells are exposed to mechanical forces from changes in tissue movements, deformations, and stiffness. In order for tissues to develop and function correctly, there must be mechanisms that couple cell signaling with the mechanical environment. Indeed, there is evidence that changes in cell morphology can impact

on gene expression (*Alam et al., 2016*; *Guilluy et al., 2014*), but this has been little explored in the context of developmental signaling.

Notch is a key developmental signaling pathway that transmits information between cells in contact. Prior to ligand binding, the ADAM10 cleavage site in Notch is hidden by the negative regulatory region (NRR). To reveal this site, ligand binding exerts a force on the receptor, leading to a displacement of the NRR (*Gordon et al., 2007*; *Gordon et al., 2015*). A second cleavage by γ-secretase then releases the intracellular domain, NICD, which forms a complex with a DNA-binding transcription factor and a co-activator to regulate transcription of target genes. In many developmental processes, Notch activation occurs contemporaneously with morphological changes which could modulate pathway activity so that signaling and tissue rearrangements are coordinated (*Paolini et al., 2021*; *Han et al., 2021*; *Engel-Pizcueta and Pujades, 2021*; *Lloyd-Lewis et al., 2019*). For example, cell shape or tension changes in the neighboring cells could impact on the forces exerted on the receptor to alter the amount of cleavage (*Shaya et al., 2017*). Alternatively, mechanical forces could alter the transcriptional output from Notch activation, by changing transport through nuclear pores or altering chromatin compaction (*Boumendil et al., 2019*; *Gozalo et al., 2020*).

To distinguish whether morphological events exert an influence on signaling, it is important to monitor the outputs in real-time when the changes are occurring. We have been investigating the onset of Notch activity in the *Drosophila* embryo using the MS2-MCP system to monitor the transcriptional response live. Notch activity initiates in a stripe of cells flanking the mesoderm, the mesectoderm (MSE), in the last cycle prior to gastrulation (nuclear cycle 14, nc14), and remains active throughout gastrulation. These cells divide after gastrulation, change their polarity and form a boundary that separates the flanking neuroectoderm (*Yu et al., 2021*). They subsequently form the midline of the central nervous system, giving rise to specific neuronal and glial progeny (*Wheeler et al., 2008*) under the control of the Notch regulated *single-minded* gene, which is essential for cell division after gastrulation, acquisition of their midline fate and proper axonal connectivity (*Nambu et al., 1990*; *Nambu et al., 1991*; *Hummel et al., 1999*). Defects in the establishment of the mesectoderm boundary result in mixing of the left and right neuroectoderm as well as later effects on midline development (*Yu et al., 2021*). A notable feature of the transcriptional profiles from two Notch-responsive enhancers, *sim* and *m5/m8*, is that they undergo a transition approximately 50 min into nc14, when transcription levels increase approximately twofold (*Falo-Sanjuan et al., 2019*). This transition occurs as the embryo undergoes gastrulation, making it plausible that mechanical forces or tissue reorganization at this stage are responsible for the increase in transcription, and may contribute to the integrity of boundary formation (*Nambu et al., 1990*; *Yu et al., 2021*).

Gastrulation initiates toward the end of nc14, approximately 3 hr post fertilization. During this process, the apical surface of the most ventral subset of mesoderm (ME) cells constricts and cells shorten along their apico-basal axis (*Leptin and Grunewald, 1990*; *Sweeton et al., 1991*). This occurs in response to apical re-localization of Myosin II (MyoII), which is controlled by a GPCR (G-protein coupled receptor) cascade (*Sweeton et al., 1991*; *Manning et al., 2013*; *Parks and Wieschaus, 1991*; *Barrett et al., 1997*; *Kölsch et al., 2007*; *Dawes-Hoang et al., 2005*). As a consequence, the 'ventral furrow' is formed, which invaginates bringing the remainder of the mesoderm cells with it (*Leptin and Grunewald, 1990*; *Sweeton et al., 1991*). Studies of the forces generated indicate that, although the force to invaginate the furrow is produced autonomously in the mesoderm by pulses of acto-myosin contractions (*Martin et al., 2009*), the mechanical properties of tissues adjacent to it, such as the MSE and neuroectoderm (NE), also change and may be important to allow invagination to occur (*Rauzi et al., 2015*).

We set out to investigate whether there is a causal relationship between morphological events occurring at gastrulation and the change in levels of Notch-dependent transcription, using live imaging. Strikingly, we found a strong correlation between the start of mesoderm invagination and the time at which transcription levels increased. This change in transcription was delayed or absent when gastrulation was perturbed using different genetic mutations and manipulations. A Notch-independent enhancer exhibited similar albeit more modest effect, suggesting that the mechanism involves a more general feature. Furthermore, the Notch intracellular fragment, NICD, was also subject to similar regulation. The results indicate therefore that the mechanical context has a significant impact on the transcriptional outcome of Notch signaling but argue that this operates downstream of receptor activation and also affects other developmental enhancer(s). This type of coordination between tissue

forces, developmental signaling and nuclear activity is thus likely to be of general importance for shaping tissues and organs as they form.

## Results

### Notch-dependent transcription increases at gastrulation

During early *Drosophila* embryogenesis, Notch is active in the mesectoderm (MSE), a stripe of cells that border the mesoderm (*Nambu et al., 1990*; *Morel and Schweisguth, 2000*; *Morel et al., 2003*). Activity commences midway through nc14 and continues during gastrulation, persisting in most MSE cells until they divide to form the precursors of the midline (*Martín-Bermudo et al., 1995*; *Cowden and Levine, 2002*; *Falo-Sanjuan et al., 2019*). Measuring transcription live, using the MS2 system, reveals that there is a striking increase in the mean levels of transcription produced by Notch-responsive enhancers during gastrulation. This increase occurs approximately 20 min after transcription initiates, which corresponds to 50 min after the start of nuclear cycle 14 (nc14). At this point, the transcription levels from two distinct Notch-responsive mesectodermal enhancers, *m5/m8* and *sim* (*Zinzen et al., 2006*; *Hong et al., 2013*), almost double in magnitude (*Figure 1AB*, *Falo-Sanjuan et al., 2019*). The change in magnitude occurs with different promoter combinations and with reporters inserted into different genomic positions (*Figure 1—figure supplement 1AC*, *Falo-Sanjuan et al., 2019*), arguing it is a feature of the transcriptional response driven by the Notch-responsive enhancers. However, there was no coordinated transition in levels when the profiles from all cells were aligned by activity onset in the MSE (*Figure 1—figure supplement 2B*), arguing that the increase is not dependent on the length of time that the enhancers have been active and thus may be related to a specific developmental event/time.

The observed increase in the mean transcriptional activity was not due to an increase in the number of active cells (*Figure 1D*), but rather to a change in the profiles within individual nuclei. This differed in magnitude from nucleus to nucleus (*Figure 1—figure supplement 2*) . Thus, when *m5/m8* transcription profiles from individual nuclei were analyzed, a clear, circa twofold, transition in levels was detected in 30–45% of the MSE nuclei (*Figure 1C*). Others did not show such a major increase in levels, but several still manifest an inflection in the bursting at the equivalent time (*Figure 1C*).

The transition in levels occurs whilst the embryo undergoes gastrulation (*Video 1*), a process that starts midway through nc14 and lasts approximately 20 min. Gastrulation begins when the most ventral mesodermal cells constrict their apical surface to initiate ME invagination and results in the convergence of the two MSE stripes at the midline (*Figure 1GF*, *Leptin and Grunewald, 1990*; *Sweeton et al., 1991*). Given the large scale morphogenetic movements involved, it is possible that these influence the transcriptional activity. We therefore asked whether any of the changes that take place at gastrulation are correlated with the transition in transcription from the Notch-responsive enhancers, by analyzing all the time-course data previously obtained from Notch-responsive reporters (combining different enhancers, promoters and landing sites, n=55 videos, *Falo-Sanjuan et al., 2019*). On an embryo-by-embryo basis, we examined the relationship between the transcription transition-point and three different features during gastrulation: start of apical constriction, start of mesoderm invagination (defined as when mesectoderm cells first start to move ventrally) and the end of gastrulation (when all mesoderm cells have invaginated). Of the three, the highest correlation (both in terms of $R^2$ and both events occurring at similar timepoints) was with the start of mesoderm invagination (*Figure 1G*). This increase was not due to changes in the Z position of the nuclei that occurred during this period. The extent of nuclear movement with respect to the plane of imaging depends on the precise orientation of the embryo. Taking advantage of this variation between embryos, we measured the change in Z position for each in relation to the fold increase in transcription levels and found no correlation ($R^2$ coefficient of 0, *Figure 1—figure supplement 1E*). Nevertheless, because there is a temporal correlation with gastrulation, a plausible model is that the tissue level morphogenetic movements from gastrulation are responsible for the increase in transcription levels.

### Events at gastrulation modulate Notch-dependent transcription

To investigate whether morphological changes at gastrulation bring about the transition in Notch-dependent transcription, we used a combination of approaches to disrupt gastrulation while live-imaging transcription from *m5/m8* [II], an insertion of the reporter on the second chromosome that

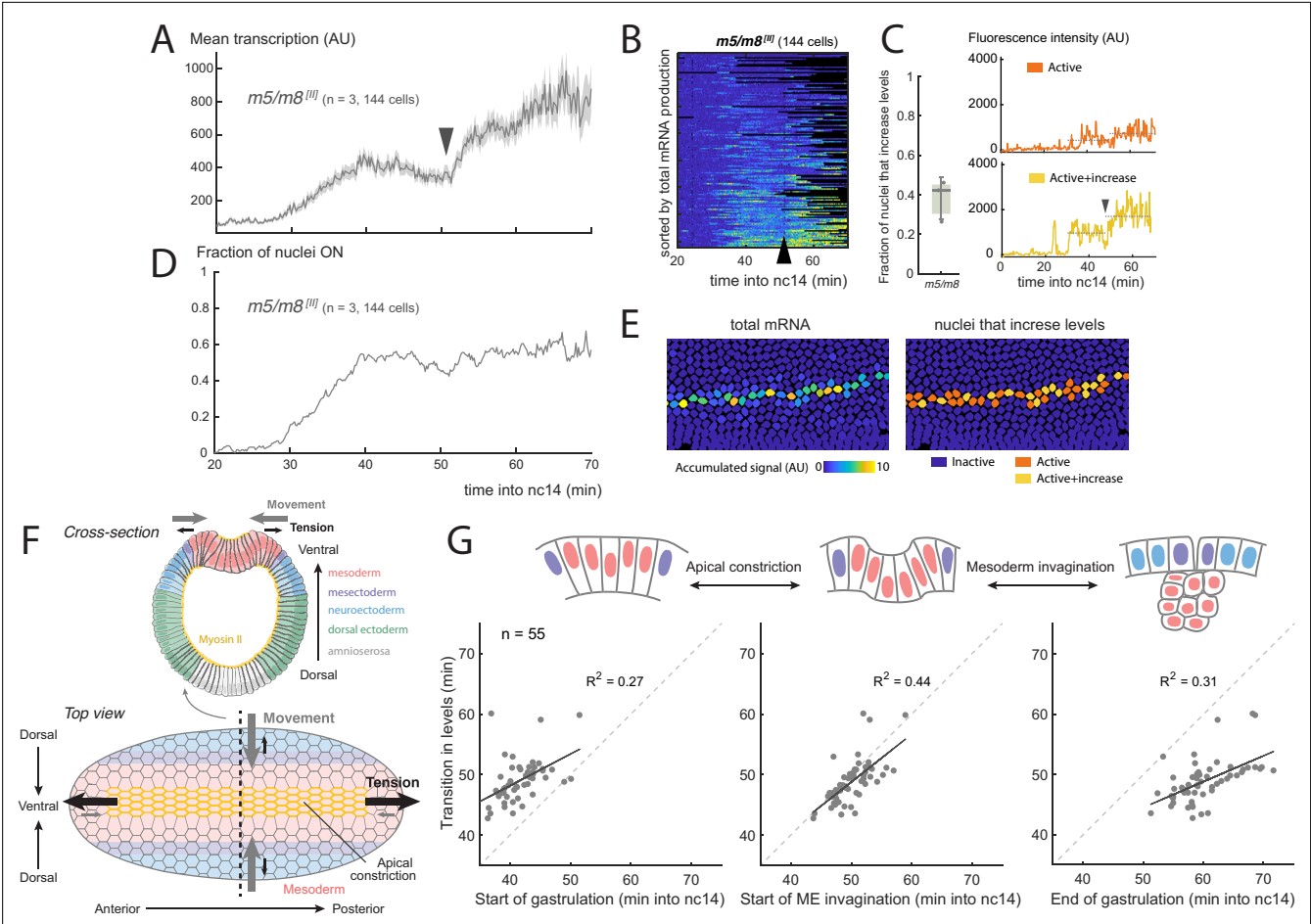

**Figure 1.** Increase in Notch-dependent transcription occurs at gastrulation. (**A**) Mean profile of activity of the Notch-responsive *m5/m8* [II] enhancer during nc14. (**B**) Heatmap showing *m5/m8* [II] activity in all MSE cells over time, arrowhead indicates the transition point. (**C**) Proportion of active cells in each embryo that increase levels of *m5/m8* [II] transcription at gastrulation (left; median, Q1/Q3 and SD shown) and examples of individual *m5/m8* [II] transcription traces with ~2x increase (yellow) or inflection only (orange). Dotted lines indicate mean transcription levels in the 15 min before and after gastrulation. (**D**) Fraction of tracked nuclei that are actively transcribing during nc14. (**E**) Still frame showing tracked nuclei color-coded for the total transcription produced by each nuclei (left) and by whether they exhibit a ~2x increase in levels (right). (**F**) Schematic drawing of the forces generated during gastrulation. A cross-section with the different regions along the dorso-ventral axis (top) and top view from the ventral side (bottom) are shown. The most ventral part of the mesoderm (red-shaded) localizes Myosin II (yellow) apically, which generates apical constrictions and leads to tension generated towards the anterior and posterior ends of the embryo, producing ventral movement from the mesoderm, mesectoderm and neuroectoderm cells. (**G**) Correlation between the time at which mean levels of transcription increase in an embryo and the timing of three events during gastrulation. Each dot indicates an embryo from a collection of different Notch-responsive enhancers, promoters and landing sites (n=55 videos). Panels A-G were obtained by re-analyzing data from *Falo-Sanjuan et al., 2019*. Panel F has been adapted from Figure 6B from *Martin, 2020* and Figure 1 from *Leptin, 1999*.

The online version of this article includes the following figure supplement(s) for figure 1:

**Figure supplement 1.** Increase in Notch-dependent transcription at the time of gastrulation.

**Figure supplement 2.** Quantification of gastrulation progression and changes in transcription levels.

responds robustly to Notch activity (*Falo-Sanjuan et al., 2019*). Gastrulation is coordinated by a signaling cascade that controls Myosin contractility, which in turn produces apical constriction of mesoderm cells to drive invagination (*Figure 2A*; *Dawes-Hoang et al., 2005*; *Kölsch et al., 2007*). First, we performed germline RNAi knockdowns (KD) to eliminate maternally encoded proteins that act in the signaling cascade, namely α-Catenin (α-Cat), a key component of Adherens Junctions (AJ), Concertina (Cta), and RhoGEF2, which are the Gα and GEF (Guanine nucleotide Exchange Factor) of the signaling cascade (*Parks and Wieschaus, 1991*; *Barrett et al., 1997*; *Figure 2A*). Apical relocalization of Armadillo (an AJ component) and RhoGEF2 is required for gastrulation (*Kölsch et al.,*

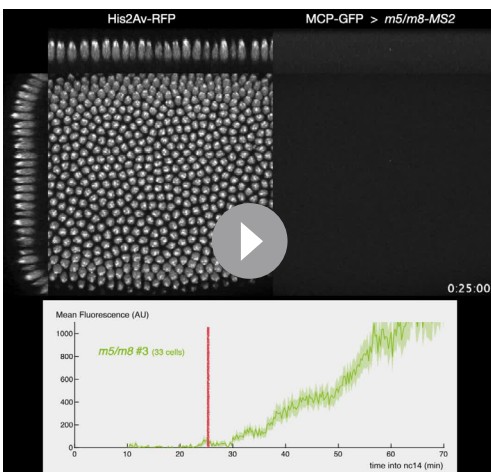

**Video 1.** Activity of *m5/m8* during gastrulation. Video showing His2Av-RFP channel (orthogonal view, left ; and maximum intensity projection, center) and MCP-GFP channel with transcription directed by *m5/m8* [II] (maximum intensity projection, right) in control embryos. The top row shows orthogonal view in the His2Av-RFP channel and maximum Y projection in the MCP-GFP channel. 0.36 µm/px XY resolution, 29x1µm slices and time resolution of 15s/frame. Anterior to the left; embryo imaged from the ventral side. Time indicates minutes from the beginning of nc14. Bottom plot shows mean transcription produced by *m5/m8* [II] in this embryo, synchronized with the video to show the increase in activity occurs as the embryo undergoes gastrulation.

https://elifesciences.org/articles/73656/figures#video1

*2007*) and Armadillo is required for apical myosin relocalization (*Dawes-Hoang et al., 2005*). We used *α-Cat RNAi* rather than *Arm RNAi* because it produced a much stronger KD. RNAi depletion produced morphological phenotypes consistent with those previously described, except in the case of RhoGEF2 (*Parks and Wieschaus, 1991*; *Barrett et al., 1997*; *Dawes-Hoang et al., 2005*), where the depleted embryos were viable and lacked obvious gross morphological defects, such as extra folds, suggesting a lower knock-down efficiency (*Figure 2—figure supplement 1A*).

Of the three tested, α-Cat KD led to the most severe disruption of gastrulation; mesoderm cells failed to invaginate and divided externally (*Figure 2—figure supplement 1A-C*, *Video 2*). Strikingly, no increase in *m5/m8* [II]-directed transcription occurred in these embryos. Instead, the mean levels decreased after the initial activation, at the time when the increase in levels normally occurs, and then plateaued at a lower level (*Figure 2B*, *Figure 2—figure supplement 2B*). This was because few individual nuclei exhibited any increase in levels in these embryos (*Figure 2—figure supplement 2A*). Other reporters that also exhibited increased levels during gastrulation, namely *sim* [II] and *m5/m8* [III], were similarly affected by α-Cat depletion (*Figure 2—figure supplement 3*). We note the mean levels of transcription from the *m5/m8* [II] reporter were not significantly reduced prior to gastrulation by α-Cat depletion, unlike those from *m5/m8* [III] and the endogenous *E(spl)m8-HLH* gene that both showed decreased levels also during cellularization (*Falo-Sanjuan and Bray, 2021*). The difference likely arises because this *m5/m8* [II] insertion achieves maximal transcription at lower levels of Notch activity, based on its response in different contexts, and that α-Cat depletion reduces, but does not fully compromise, Notch activity (*Falo-Sanjuan and Bray, 2021*).

Depletion of either Cta or RhoGEF2 resulted in a slowing of gastrulation. Cta depletion also caused variable delays in the start of mesoderm invagination (*Figure 2—figure supplement 2BC*; *Video 3*). In neither genotype was there a normal increase in transcription at the time of gastrulation. The mean levels remained similar to those prior to gastrulation and, on an individual nucleus basis, very few exhibited any marked change in levels (*Figure 2D*, *Figure 2—figure supplement 2CD*). Although there was no clear increase detected, the profiles retained an inflection point, when there was a transition between the early 'peak' levels and the later activity, manifest as a small dip in levels between the two phases. To assess whether this transition point was related to gastrulation events, the time when the transition occurred was plotted against the mile-stones for each embryo individually. The strongest correlation was with the start of mesoderm invagination (coefficient of $R^2 = 0.7$, *Figure 2C*), as it was for the increase in levels that occurs in wild-type embryos.

We next evaluated the consequences from mutations affecting the zygotically required gene *folded gastrulation* (*fog*), which encodes the ligand for the GPCR in the cascade regulating gastrulation (*Costa et al., 1994*; *Dawes-Hoang et al., 2005*). *fog*- hemizygous embryos exhibited a delayed start of mesoderm invagination and slowed gastrulation overall (*Figure 2—figure supplement 1BC Video 2*). In these embryos, levels of transcription increased, although less than in normal embryos (*Figure 2F*, *Figure 2—figure supplement 2EF*). Notably, there was a significant 10 min delay in the time at which levels increased, from approximately 50 min to 60 min into nc14 (*Figure 2F*) . This

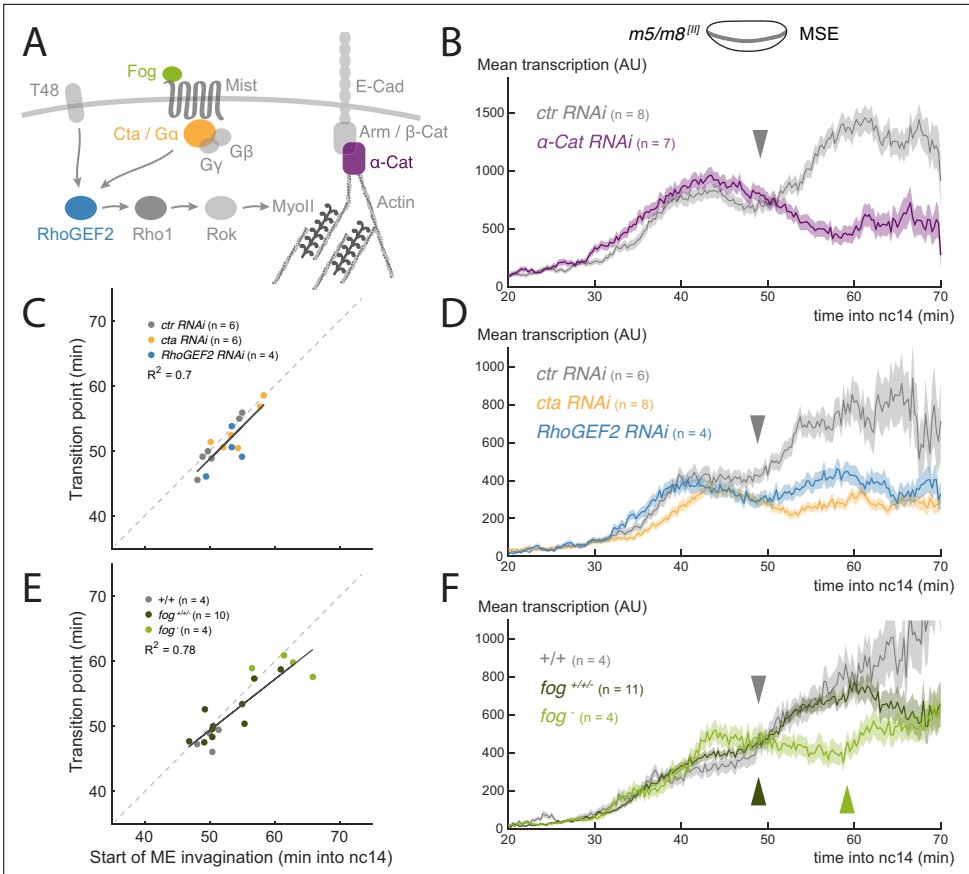

**Figure 2.** Disruption of gastrulation correlates with changes in the transition in Notch-dependent transcription.
(**A**) Simplified scheme of the signaling cascade that controls MyoII contractility during *Drosophila* gastrulation.
(**B**) Mean profile of *m5/m8* [II] activity in α-*Cat RNAi* embryos compared to control embryos. (**C**) Correlation between the start of invagination and transition in levels of transcription in each embryo, in *cta*, *RhoGEF2* and *control RNAi* embryos. (**D**) Mean profile of *m5/m8* [II] activity in *cta*, *RhoGEF2* and *control RNAi* embryos. (**E**) Correlation between the start of invagination and transition in levels of transcription in each embryo, in *fog* mutant embryos compared to control embryos and other non-*fog* hemizygous embryos obtained from the same cross. (**F**) Mean profile of *m5/m8* activity in *fog* mutant embryos compared to control embryos and other non-*fog* hemizygous embryos obtained from the same cross. The transition in levels is delayed approximately 10 min in *fog* mutants (arrowheads). Mean transcription profiles show mean and SEM (shaded area) of MS2 fluorescent traces from all cells combined from multiple embryos (n embryo numbers indicated in each). $R^2$ coefficients are calculated after pooling all points shown in the same plot together. The transition point was only considered when a clear change in mean levels of transcription in an individual embryo could be observed, therefore the number of points in **C** and **E** could be smaller than the total number of embryos collected for each condition.

The online version of this article includes the following figure supplement(s) for figure 2:

**Figure supplement 1.** Genetic disruption of gastrulation.

**Figure supplement 2.** Effects of genetic manipulations to gastrulation on the increase in *m5/m8* [II] transcription.

**Figure supplement 3.** Gastrulation also influences other Notch-responsive enhancers.

**Figure supplement 4.** Other genetic manipulations do not alter transcription profiles during gastrulation.

time-point also correlated well ($R^2$=0.78) with the start of mesoderm invagination when analyzed on an embryo-by embryo basis (*Figure 2E*) , similar to the transition-point in the RNAi depleted embryos.

We note other genetic manipulations that do not affect gastrulation do not alter the increase of *m5/m8* [II] activity at the time of gastrulation (*Figure 2—figure supplement 4*), although some affect other features of the response, such as an overall reduction in mean levels. This further supports that the observed effects are specific to gastrulation and not an indirect consequence of the manipulations performed.

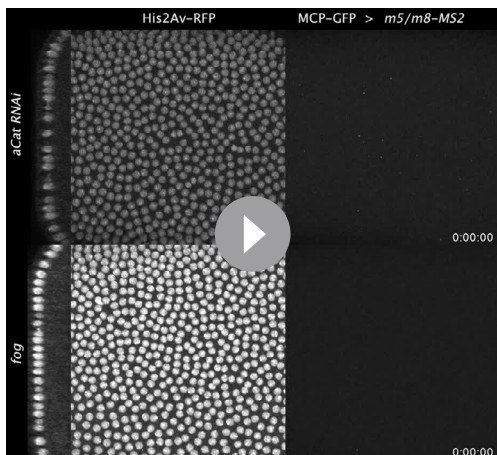

**Video 2.** Effects on gastrulation of *α-Cat RNAi* and *fog* mutant background. Videos showing His2Av-RFP channel (orthogonal view, left; and maximum intensity projection, center) and MCP-GFP channel with transcription directed by *m5/m8* [II] (maximum intensity projection, right) in *α-Cat RNAi* (top) and *fog*⁻ (bottom) embryos. 0.36 µm/px XY resolution, 27x1µm (*α-Cat RNAi*) and 29 × 1µm (*fog*) slices and time resolution of 15 s/frame. Anterior to the left; embryo imaged from the ventral side. Time indicates minutes from the beginning of nc14.

https://elifesciences.org/articles/73656/figures#video2

causal step.

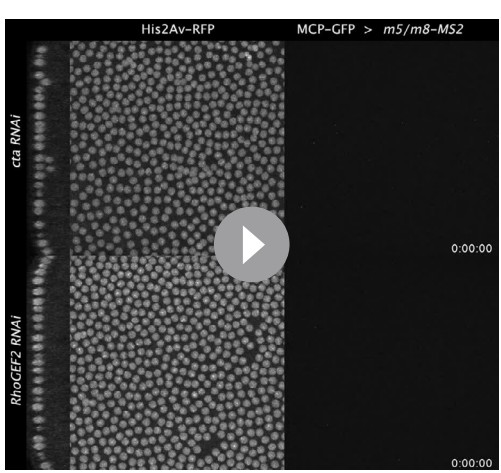

**Video 3.** Effects on gastrulation of *cta* and *RhoGEF2 RNAi*. Videos showing His2Av-RFP channel (orthogonal view, left; and maximum intensity projection, center) and MCP-GFP channel with transcription directed by *m5/m8* [II] (maximum intensity projection, right) in *cta* (top) and *RhoGEF2 RNAi* (bottom) embryos. 0.36 µm/px XY resolution, 29x1µm slices and time resolution of 15s/frame. Anterior to the left; embryo imaged from the ventral side. Time indicates minutes from the beginning of nc14.

https://elifesciences.org/articles/73656/figures#video3

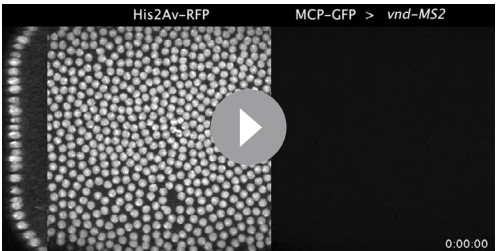

**Video 4.** Expression of *vnd* in control embryos. Videos showing His2Av-RFP channel (orthogonal view, left; and maximum intensity projection, center) and MCP-GFP channel with transcription directed by *vnd* (maximum intensity projection, right) in control embryos. 0.36 µm/px XY resolution, 29x1µm slices and time resolution of 15s/frame. Anterior to the left; embryo imaged from the ventral side. Time indicates minutes from the beginning of nc14.

https://elifesciences.org/articles/73656/figures#video4

Based on these genetic experiments, the increase in Notch-dependent transcription levels is perturbed or delayed in conditions where gastrulation is blocked or slowed down. We hypothesize therefore that normal 'fast' gastrulation is required for the increase in levels and, since we observe a consistent correlation between the start of mesoderm invagination and the transition in transcription levels, it is likely that this is the

## Gastrulation also modulates Notch-independent transcription

There are several different types of mechanisms that could explain a causal link between gastrulation and Notch-dependent transcription levels. As Notch activation involves a pulling force from Delta in adjacent cells (*Gordon et al., 2015*), one model is that increased cell surface tensions from the morphogenetic movements at gastrulation lead to increased Notch cleavage and NICD release. An alternative model is that the forces bring about a change in nuclear properties that consequently impact on transcription. For example, these could include changes in nuclear import, chromatin reorganization or chromatin mobility.

To distinguish these possibilities we took two approaches. First, we generated a transcriptional reporter using a Notch-independent enhancer from the *ventral nervous system defective* (*vnd*) gene. Unlike many of the other embryonic enhancers studied to date, the *vnd early embryonic enhancer* (*vndEEE*) is reported to be active throughout nc14 and to drive expression in a band of cells that overlap the MSE (*Stathopoulos*

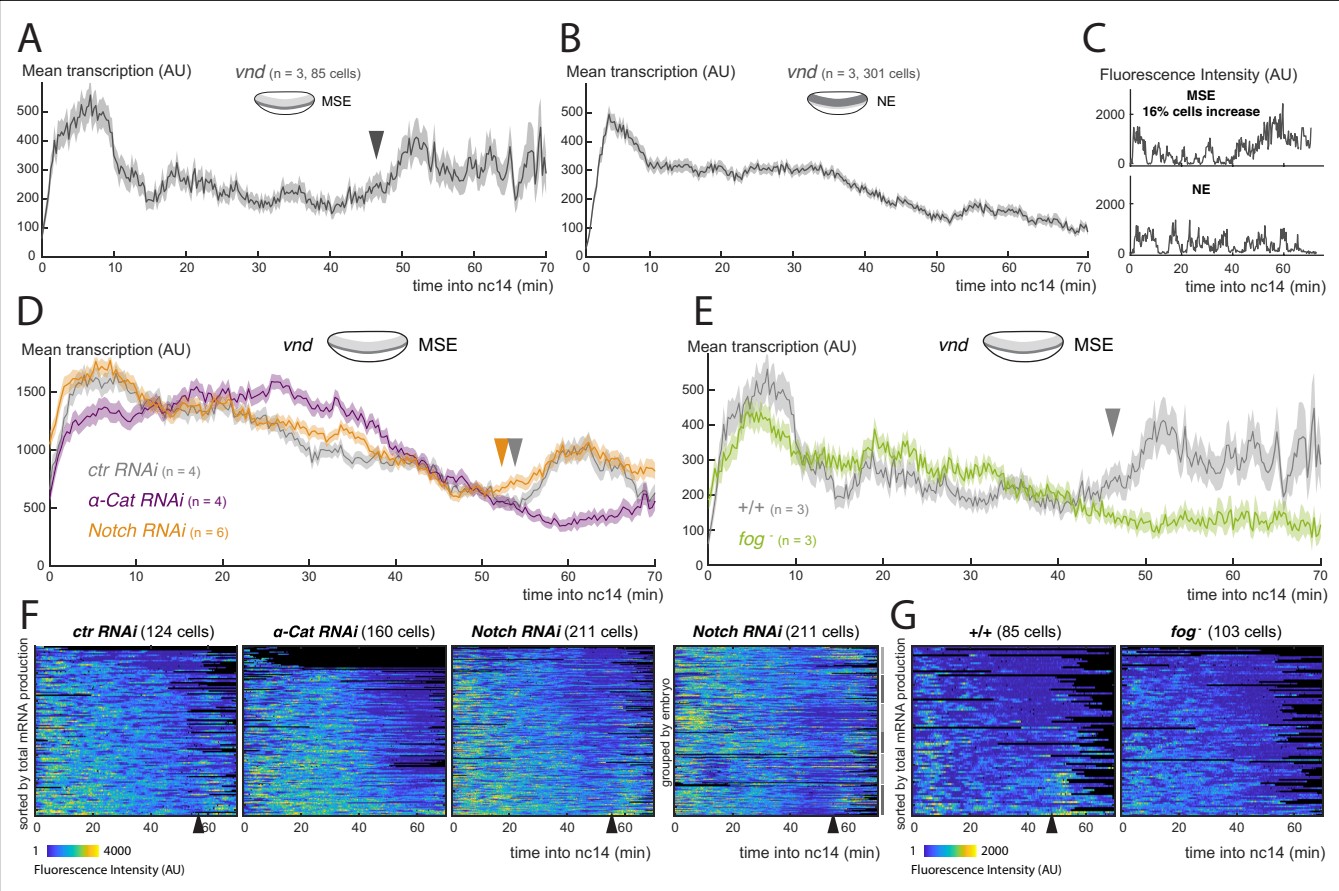

**Figure 3.** Activity of the Notch-independent *vnd* enhancer is modulated by gastrulation. (**A**) Mean profile of transcription from *vnd* in MSE nuclei. (**B**) Mean profile of transcription of *vnd* in NE nuclei. (**C**) Examples of *vnd* transcription traces from MSE and NE, with (upper) and without (lower) a marked increase at the time of gastrulation. (**D**) Mean profile of transcription from *vnd* in MSE nuclei in α-Cat and *Notch RNAi* compared to control embryos. (**E**) Mean profile of transcription from *vnd* in MSE nuclei in *fog* mutant embryos compared to control embryos. (**F**) Heatmaps showing *vnd* activity in all MSE cells over time in α-Cat, *Notch* and *control RNAi* embryos, sorted by total mRNA production (first 3 plots) and *Notch RNAi* grouped by embryo. (**G**) Heatmaps showing *vnd* activity in all MSE cells over time in *fog* embryos compared to controls, sorted by total mRNA production. Mean transcription profiles show mean and SEM (shaded area) of MS2 fluorescent traces from all cells combined from multiple embryos (n embryo numbers indicated in each). Arrowheads indicate increase in the mean levels of transcription.

The online version of this article includes the following figure supplement(s) for figure 3:

**Figure supplement 1.** *Notch RNAi* strongly reduces Notch-dependent transcription.

*et al., 2002*). As predicted, a new MS2 reporter containing this enhancer, referred to here as *vnd*, was active from the beginning of nc14 throughout gastrulation and recapitulated the spatial pattern of *vnd* (*Video 4*). After a peak of transcription at the start of nc14, mean levels of *vnd* transcription from all active nuclei exhibited no increase in levels at gastrulation. Indeed, the overall mean decreased at this time. However, when nuclei were separated according to their expression domain, an increase in mean levels was detectable specifically in MSE nuclei at around 50 min into nc14, similar to the *m5/m8* enhancer (*Figure 3A*). No increase was detected in the other, NE domain (*Figure 3B*). The proportion of nuclei that exhibited a clear increase in levels when considered individually was, however, considerably lower than for *m5/m8* (16% of MSE nuclei and 2% of NE nuclei, *Figure 3C*). These results suggest that the activity of Notch-independent as well as Notch-dependent enhancers are affected at the time of gastrulation, albeit the effect is more modest for Notch-independent activity, and that this property is limited to the mesectodermal cells.

We next investigated whether the increase in *vnd* activity in MSE nuclei was also linked to gastrulation, by measuring transcription in α-Cat depleted and *fog* mutant embryos. Neither genotype showed an increase in transcription in MSE cells (*Figure 3D-G*), instead the levels remained at a plateau similar to that in the NE nuclei (*Figure 3B*). Thus, it appears that *vnd* transcription in MSE

nuclei is also augmented due to gastrulation, in a similar manner to *m5/m8*-directed transcription. As there is no evidence that *vnd* is regulated by Notch (*Markstein et al., 2004*), this implies that gastrulation exerts effects on transcription independent of any effects on Notch activation. To verify that the modulation of *vnd* is not Notch dependent, we used an RNAi line to deplete Notch levels. As predicted, this treatment greatly reduced transcription from *m5/m8* [II] (*Figure 3—figure supplement 1A,B*). In contrast, there was no change in the timing of either the transition or the increase in levels from the *vnd* enhancer in any of the *Notch RNAi* embryos ( *Figure 3DF*). Together the results indicate that gastrulation also modulates the activity of a Notch-independent *vnd* enhancer, suggesting that a general, rather than a Notch-specific, mechanism is involved.

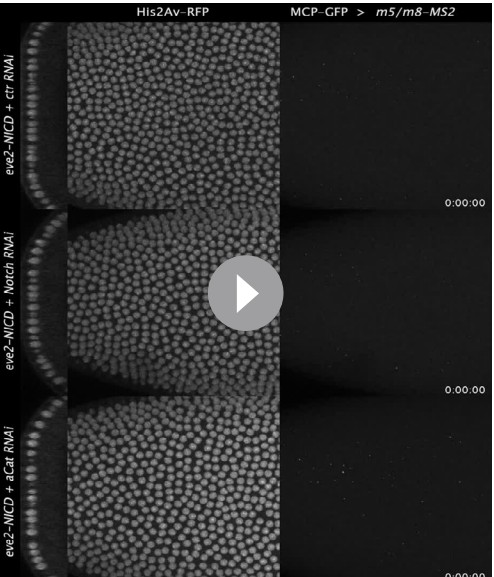

**Video 5.** Expression of *m5/m8* [II] upon *eve2-NICD* expression in control, *Notch* and *α-Cat RNAi* embryos. Videos showing His2Av-RFP channel (orthogonal view, left ; and maximum intensity projection, center) and MCP-GFP channel with transcription directed by *m5/m8* [II] (maximum intensity projection, right) in control (top), *Notch* (middle) and *α-Cat RNAi* (bottom) embryos. 0.36 μm/px XY resolution, 32x1μm slices and time resolution of 20s/frame. Anterior to the left; embryo imaged from the ventral side. Time indicates minutes from the beginning of nc14.
https://elifesciences.org/articles/73656/figures#video5

## Regulation of Notch-dependent transcription by gastrulation occurs downstream of pathway activation

The results with the *vnd* enhancer suggest that the gastrulation-induced changes in *m5/m8* transcription do not arise from increased Notch cleavage. As a second approach to investigate at which level of the pathway this modulation occurs, we examined whether gastrulation exerted any effects on transcription levels when the intracellular fragment NICD was expressed ectopically. To do so, we used an *eve2* transgene to direct expression of NICD in an orthogonal stripe overlapping the MSE, which is sufficient to drive ectopic transcription from *m5/m8* (*Kosman and Small, 1997*; *Cowden and Levine, 2002*; *Falo-Sanjuan et al., 2019*). The mean profile of *m5/m8* [II] transcription in MSE nuclei that were exposed to NICD had two peaks, the first shortly after the onset of NICD expression and the second during gastrulation. The transition between the two, the 'trough', slightly preceded the onset of mesoderm invagination. Thus, the second 'peak' was initiated at the start of mesoderm invagination, characteristic of the gastrulation-induced increase in transcription in normal conditions. As MSE nuclei will be exposed to endogenous Notch signaling as well as the ectopic NICD, we also examined the profiles in the bordering NE nuclei. These exhibited a similar trough in levels prior to gastrulation, although the subsequent activity did not achieve the same levels as in the MSE nuclei (*Figure 4—figure supplement 1B*) . Together, these data support the model that gastrulation has an effect on *m5/m8* transcription that is independent of any influence on Notch cleavage.

To further verify that gastrulation-dependent changes occur independent from Notch cleavage, we monitored the transition in transcription levels elicited by NICD in embryos where the endogenous Notch was depleted by RNAi. No transcription was detected outside the domain of the *eve2* stripe (*Figure 4—figure supplement 1A*, *Video 5*), confirming that Notch depletion was successful. Strikingly, MSE nuclei within the stripe where NICD was expressed exhibited the same profile of activity in *Notch RNAi* depleted embryos to those from control embryos with intact Notch. In particular, the levels of transcription increased at the time of gastrulation and to the same degree (*Figure 4AC*). Thus, the transition in transcription levels at gastrulation occurs even in the absence of full-length Notch, arguing the effects are downstream of receptor cleavage.

To investigate whether the transition in transcription that occurs in the presence of NICD is directly related to gastrulation, we examined it in embryos where gastrulation was disrupted by α-Cat depletion or the *fog* allele, as above. In the α-Cat depleted embryos, where gastrulation was completely

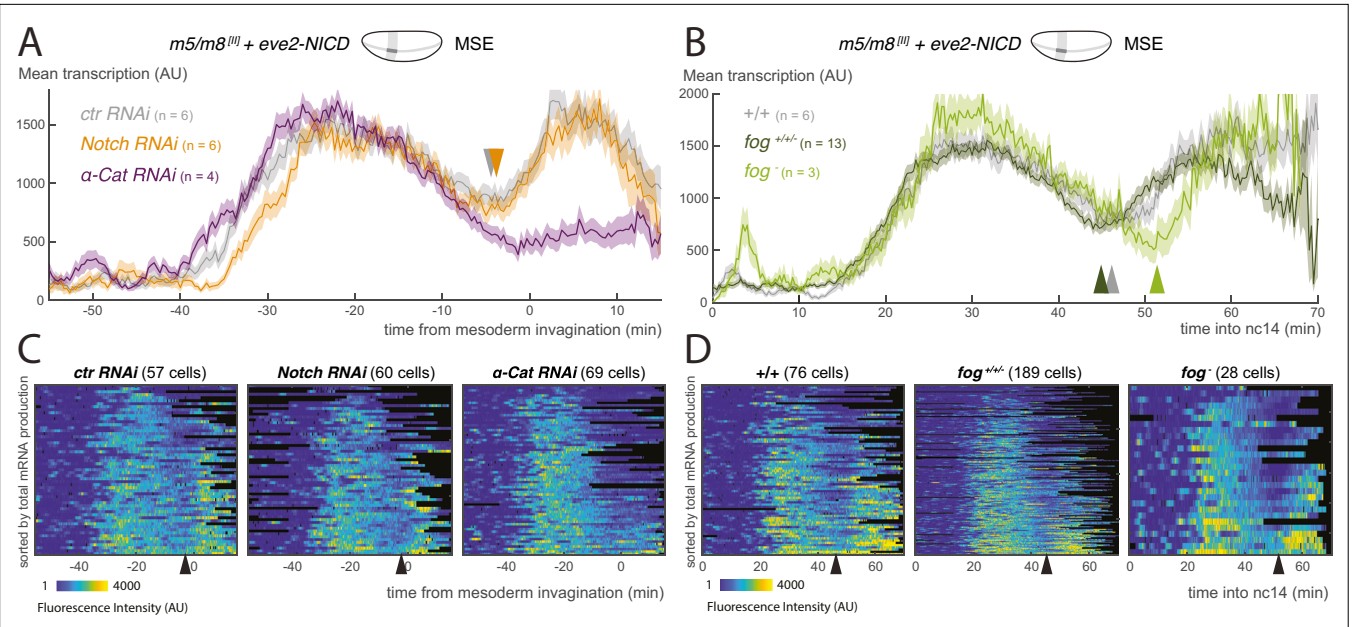

**Figure 4.** Modulation of Notch-dependent transcription occurs downstream of pathway activation. (**A**) Mean profile of transcription from *m5/m8* [II] in MSE nuclei, aligned by the time of ME invagination, in α-*Cat* and *Notch RNAi* compared to control embryos in the presence of ectopic NICD (produced by *eve2-NICD*). (**B**) Mean profile of *m5/m8* [II] transcription in MSE nuclei with ectopic NICD in homozygous *fog* mutants compared to control or heterozygous embryos from the same cross (*fog*[+/+/-]). (**C**) Heatmaps showing *m5/m8* [II] activity in MSE cells within *eve2-NICD* domain, aligned by the time of ME invagination, in α-*Cat*, *Notch* and *control RNAi* embryos. (**D**) Heatmaps showing *m5/m8* [II] activity in MSE cells within *eve2-NICD* domain over time in *fog*[-] embryos compared to control or heterozygous embryos obtained from the same cross (*fog*[+/+/-]). Mean transcription profiles show mean and SEM (shaded area) of MS2 fluorescent traces from all cells combined from multiple embryos (n embryo numbers indicated in each). Arrowheads indicate an increase in the mean levels of transcription.

The online version of this article includes the following figure supplement(s) for figure 4:

**Figure supplement 1.** Modulation of transcription occurs downstream of pathway activation.

disrupted, there was no second 'peak' of transcription from *m5/m8* [II]. Instead the levels remained at low levels from the time when mesoderm invagination would normally occur (*Figure 4AC*). *fog* mutant

embryos, in which gastrulation was delayed and slowed down, retained a second peak of *m5/m8* [II] transcription, but its onset was delayed in both MSE and NE nuclei (*Figure 4BD*, *Figure 4—figure supplement 1BC*) . Notably, on an embryo by embryo basis, the delayed transition-point in the MSE correlated with the start of mesoderm invagination ($R^2$ coefficient of 0.62) (*Figure 4—figure supplement 1D*) .

Altogether, the results suggest that the mechanics of gastrulation have an input into transcription in the MSE, that brings about an increase in the levels of mRNA produced. Because the effects on the *m5/m8* enhancer are more marked than those on *vnd*, Notch-dependent processes may be more sensitive. Furthermore, the consequences are strongest in MSE cells, suggesting that gastrulation exerts differential impacts across the ectoderm, which might be related to the gradients of forces and cell deformation that have been measured (*Fuse et al., 2013*; *Rauzi et al., 2015*). Indeed, only a subset of mesectoderm

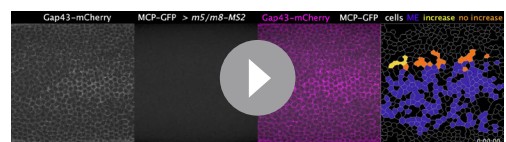

**Video 6.** Analysis of cellular shape in relation to *m5/m8* activity. From left to right, video showing: (i) membranes using the marker Gap43-mCherry (maximum intensity projection of 3 μm after surface correction), (ii) transcription from *m5/m8-MS2* [II] (maximum intensity projection of the whole stack after surface correction), (iii) overlaid Gap43-mCherry (magenta) and MCP-GFP (grey) signal, (iv) tracked video showing cell boundaries (grey), cells classified as mesoderm (purple), mesectoderm cells that increase MS2 signal during gastrulation (yellow) and mesectoderm cells that do not increase (orange). 0.36 μm/px XY resolution, 36x1μm slices and time resolution of 20 s/frame. Anterior to the left; embryo imaged from the ventral side. Time indicates minutes from the beginning of nc14.
https://elifesciences.org/articles/73656/figures#video6

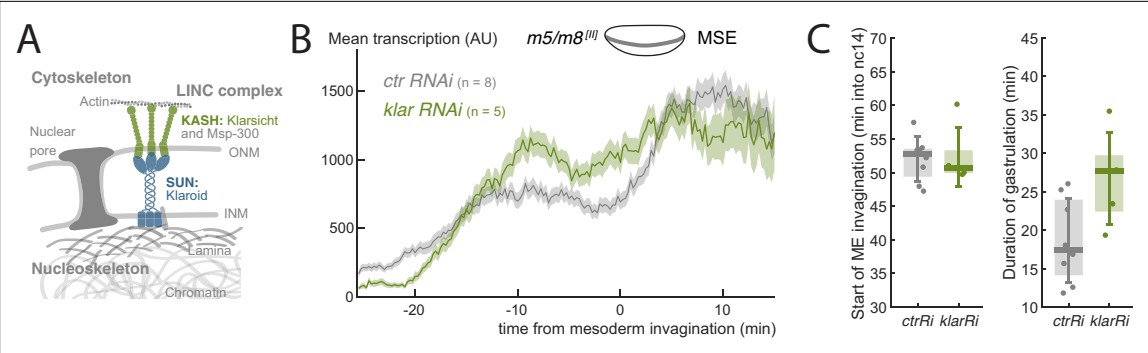

**Figure 5.** LINC complex disruption decouples gastrulation progression and changes in transcription. (**A**) Scheme depicting LINC complex that connects cytoskeleton and nucleoskeleton, *Drosophila* KASH and SUN proteins Klarsicht and Klaroid are highlighted. (**B**) Mean profile of transcription from *m5/m8*[II] in *klar RNAi* compared to control embryos. (**C**) Onset of ME invagination (left) and duration of gastrulation in *klar* and *control RNAi* embryos. Boxplots show median, Q1/Q3 quartiles and SD. Control embryos are the same as shown in *Figure 2* and *Figure 2—figure supplement 1B*.

The online version of this article includes the following figure supplement(s) for figure 5:

**Figure supplement 1.** Changes in cellular morphology during gastrulation do not correlate with cells that increase transcription levels or not.

**Figure supplement 2.** LINC complex disruption decouples gastrulation progression and changes in transcription.

cells exhibit a marked increase in transcription levels, suggesting there may be specific properties that are involved. We therefore re-analyzed videos from embryos expressing the cell membrane marker Gap43-mCherry and *m5/m8*[II] transcription (*Falo-Sanjuan and Bray, 2021*), to track cell membranes over time and measure their properties (*Figure 5—figure supplement 1A*, *Video 6*). We found no difference in the area of mesectodem cells that exhibited increased *m5/m8*[II] transcription compared to their neighbors where there was no marked increase (*Figure 5—figure supplement 2B*) nor in the eccentricity or length of their contact with mesoderm cells. A feed-forward loop between mechanical forces and Fog has been previously described, which involves inhibition of Fog endocytosis and could also be affecting Notch signaling by regulation of endocytosis (*Pouille et al., 2009*). However, as the step-change in transcription is affected by the loss of Fog and α-Cat, and also affects *vnd* and NICD-dependent transcription, the mechanism appears to involve a step subsequent to the alteration of endocytosis.

## Disruption of LINC complex alters the profile of *m5/m8* transcription

Amongst mechanisms that could be responsible for the gastrulation-related increase in transcriptional activity in MSE nuclei, one hypothesis is that forces from cell rearrangements are transmitted through the cytoskeleton to the nucleus via the Linker of Nucleoskeleton and Cytoskeleton (LINC) complex (*Figure 5A*, *Crisp et al., 2006*). In several other contexts, LINC is integral for transducing nuclear mechanical signals to confer changes in both gene expression and chromatin organization (*Hamouda et al., 2020*). We therefore set out to test the impact from disrupting components of the LINC complex on the transcription profile of *m5/m8*[II]. Of the three conditions tested, only maternal knockdown of *klarsicht* (*klar*), which encodes a KASH protein, was successful, producing embryos with strongly reduced *klar* mRNA (*Figure 5—figure supplement 2A*). Depletion of mRNA encoding SUN protein Klaroid (Koi) was unsuccessful and *Lamin C* mRNA knockdown produced no viable embryos. We therefore focused on the effects produced by *klar* mRNA depletion. Klar is known to influence lipid bodies in the embryo and *klar* mutant embryos are more transparent than controls (*Supatto et al., 2009*). Consistently, *klar RNAi* embryos exhibited higher fluorescent intensity signals (control embryos had 15.8% lower RFP and 11% lower GFP signals, *Figure 5—figure supplement 2DE*). We thus compensated the quantified MS2 signal in *klar RNAi* embryos for increased clearing by 11% (*Figure 5—figure supplement 2F*).

If the LINC complex was involved in transcription modulation, we would expect its disruption to result in similar flat transcription levels to those seen when gastrulation is perturbed. Indeed, Klar disruption did yield flatter levels of *m5/m8*[II] transcription. The early phase of transcription was slightly elevated and there was no subsequent transition or reduction in the late phase (*Figure 5B*, *Figure 5—figure supplement 2B*). As there was no distinct transition at the time of mesoderm invagination, this

suggests there is an uncoupling of the transcription profiles from the gastrulation movements caused by LINC complex disruption. However, as these embryos exhibited longer and less robust gastrulation (*Figure 5C*, *Figure 5—figure supplement 2C*) it is not possible to fully rule out that the lack of a normal increase in transcription are the result of these indirect effects on gastrulation. Nevertheless, given the key role played by Klar in linking the cytoskeleton to the nucleus, it is highly plausible that the change in profile from the *klar* mRNA knock-down occurs because the gastrulation movements have become uncoupled from transcription.

## Changes in nuclear properties at gastrulation

We set out to explore mechanisms to explain how gastrulation alters transcription, via the LINC complex. We considered two possibilities. One is that it brings about a change in nuclear import properties, as has previously been reported (*Uzer et al., 2018*; *Jahed et al., 2016*). This could involve a general change in nuclear import or a specific change in nuclear levels of key transcription factors including Twist, which is required for activity of both *m5/m8* and *vnd*. Indeed, levels of Twist are regulated by mechanical forces in some contexts (*Farge, 2003*; *Desprat et al., 2008*). A second possibility is that the force on the nucleus results in changes in chromatin compaction or accessibility, rendering the target loci more accessible to NICD complexes or to transcription machinery.

We first investigated whether there were any general changes in the nuclear import properties by analyzing the nuclear levels, and nuclear exclusion, of fluorescently-tagged molecules of different dimensions, including fluorescently-tagged dextrans of two molecular weights (70 kDa and 40 kDa), as well as several tagged variants of GFP, with and without a nuclear localization signal (nls). None of these exhibited any change in nuclear levels at the time of gastrulation, (*Figure 6—figure supplement 1AB*), including 70kDa-FITC dextran which exhibited the highest nuclear exclusion (*Figure 6—figure supplement 1A*), consistent with previous data (*Hampoelz et al., 2016*). Likewise, there were no gastrulation-related changes in nuclear levels of several core transcriptional regulators, including subunits of Mediator complex (Med4, Med17, Skd, *Figure 6—figure supplement 1C*) and the histone acetylase CBP/P300 (Nejire) (*Figure 6—figure supplement 1D*) . These results argue that there is not a general change in the permeability of nuclear pores that can account for the increase in transcription at gastrulation.

We next investigated whether nuclear levels of Twist change at the time of gastrulation. To do so, we used a Twist-LlamaTag fusion (*Bothma et al., 2018*), so that nuclear levels of newly encoded Twist could be detected via a ubiquitously expressed mCherry, without the delay from protein folding that occurs when visualizing fluorescent proteins directly (*Figure 6C*) . The measurements showed that Twist levels increase in the mesoderm throughout nc14. However, no clear change or additional increase in nuclear levels occurred in mesectoderm cells at the time of gastrulation (*Figure 6AB*, *Video 7*). These results argue that changes in Twist import/levels are not responsible for the observed effects of gastrulation on transcription and are consistent with our previous observation that mutating all Twist binding sites in *m5/m8* [II] does not prevent the increase (*Falo-Sanjuan et al., 2019*). We note also that nuclear levels of Su(H)-GFP increased continuously throughout nc14 in a similar manner to Twist, with no step-wise change at the time of gastrulation, while H-GFP remained constant (*Figure 6—figure supplement 1D*) . Overall, these observations suggest that changes in nuclear import, either general or of specific transcription factors, are not responsible for the increased transcriptional activity occurring during gastrulation.

Mesectoderm cells tilt and change shape during gastrulation (*Leptin, 1999*; *Rauzi et al., 2015*). To investigate the possibility that cellular deformation induced by gastrulation leads to changes in the nucleus we first imaged a nuclear membrane marker (Nup107-RFP) while monitoring transcription with MS2 system and quantified morphological properties of the nucleus. We found that, during gastrulation, mesoderm nuclei decrease the longest axis and increase the second (i.e. they become shorter and wider) and mesectoderm nuclei do so to a greater extent than those in the neuroectoderm but less than those in the mesoderm (*Figure 6D*, *Figure 6—figure supplement 2*). The shape-changes could explain the increased sensitivity of the mesectoderm nuclei to gastrulation, because changes in nuclear shape can bring about changes in chromatin organization or accessibility (*Guilluy et al., 2014*; *Versaevel et al., 2012*). It is challenging to quantify chromatin organization and accessibility of a small cell population in living embryos in a time-resolved manner but we used two different quantifications to probe this question. As a proxy for chromatin compaction, we measured the levels of *His2Av-RFP* at

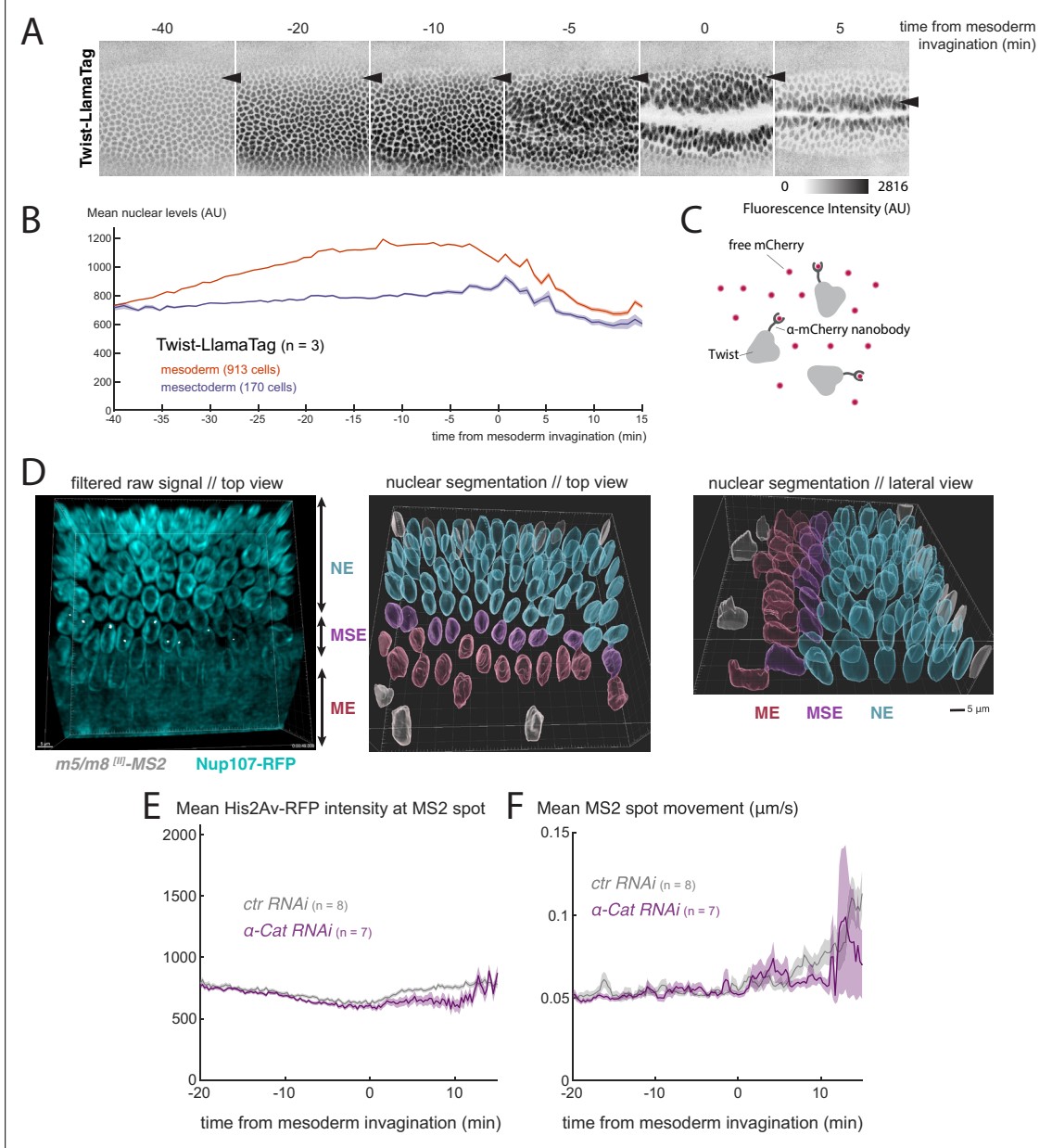

**Figure 6.** Nuclear levels of Twist and chromatin properties during gastrulation. (**A**) Still images from a video of an embryo expressing endogenous Twist fused to LlamaTag, which binds mCherry and allows to visualize nuclear levels of zygotically produced Twist (see C). Arrowhead indicates the position of the mesectoderm, note that Twist levels remain lower in these nuclei than in the central mesoderm. (**B**) Quantification of nuclear levels of Twist in mesoderm and mesectoderm during gastrulation. (**C**) LlamaTag approach: Twist is fused to an anti-mCherry nanobody (LlamaTag) that binds free and already fluorescent mCherry that has been maternally loaded in the embryo. (**D**) Still of the start of mesoderm invagination from a video of an embryo expressing the nuclear membrane marker *Nup107-RFP* (cyan) and MCP-GFP (grey) to quantify *m5/m8* [II] transcription and mark mesectoderm nuclei (left) and segmented nuclei color-coded based on the different cell populations defined from MS2 signal (ME: mesoderm, MSE: mesectoderm, NE: neuroectoderm). Center panel shows view from the top and right panel shows lateral view. (**E**) Quantification of average His2Av-RFP intensity around the MS2 spot, as a proxy of chromatin compaction, over time in *control* and α-Cat RNAi embryos. (**F**) Quantification of the average MS2 spot movement (distance moved relative to the nucleus centroid / time) over time in *control* and α-Cat RNAi embryos. **B, E and F** show mean and SEM (shaded area) from all cells combined from multiple embryos (n embryo numbers indicated in each). In F, values were smoothed over time using a median filter of 8 frames (2 min).

The online version of this article includes the following figure supplement(s) for figure 6:

**Figure supplement 1.** Nuclear import does not change during gastrulation.

**Figure supplement 2.** Quantification of nuclear size and shape during gastrulation.

**Figure supplement 3.** Quantification of chromatin properties during gastrulation.

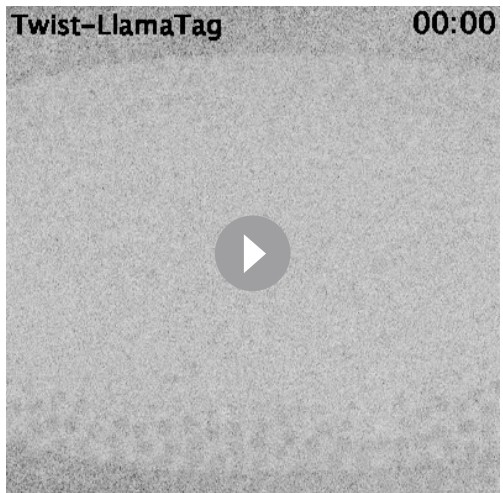

**Video 7.** Levels of nuclear Twist during nc14. Video of an embryo expressing endogenous Twist fused to a LlamaTag (anti-mCherry nanobody) that binds maternally provided and fluorescenty matured mCherry. Inverted maximum intensity projection shown. 0.36 μm/px XY resolution, 34x1μm slices and time resolution of 45s/frame. Anterior to the left; embryo imaged from the ventral side. Time indicates minutes from the beginning of nc14.
https://elifesciences.org/articles/73656/figures#video7

the site of transcription indicated by the MS2 spot. We were unable to detect any change in intensity at the time of gastrulation or across conditions (*Figure 6E*, *Figure 6—figure supplement 3A*) , suggesting there is no large scale change in chromatin density. Similarly, we used the movement of the MS2 spots to quantify relative mobility of the transcribing locus over time, because chromatin mobility is thought to be associated with its accessibility and activity (*Gu et al., 2018*). We observed that there was increased movement of the MS2 transcription puncta during gastrulation (*Figure 6F*, *Figure 6—figure supplement 3B*) , suggesting that there is an increase in the local accessibility. However, it was difficult to resolve whether this change was a key factor in the gastrulation mediated effects on transcription as the movements became highly erratic in the conditions that impacted gastrulation.

## Discussion

Morphogenetic processes integrate with programs of transcriptional regulation during animal development, but little is known about crosstalk between mechanics and transcription, particularly in a native in vivo context. Here we have investigated a role of gastrulation in modulating transcription levels in *Drosophila* embryos, where it could help to form a straight midline boundary. Using the MS2 system to quantitatively image transcription live, a clear transition in the mean levels of transcription from the Notch-responsive *m5/m8* enhancer could be detected, and to a lesser degree from the Twist and Dorsal responsive *vnd* enhancer. This transition correlated with the start of mesoderm invagination and was delayed or absent in embryos where gastrulation was perturbed by different manipulations. In conditions where there was a delay, such as in *fog* mutants, the transition in levels was correlated with a delay in mesoderm invagination. We propose that the coupling of the gastrulation movements to the transcription of key targets in the mesectoderm is important to co-ordinate the establishment of a robust and straight boundary at the midline of the embryo.

A plausible link between mechanics and Notch activity, to explain the increase in expression of Notch-regulated enhancers, would be that force is required for Notch cleavage (*Gordon et al., 2015*). By increasing membrane tension or cell contacts, the morphogenetic movements associated with gastrulation could enhance the levels of NICD released. Two observations make this scenario an unlikely explanation for the effects on transcription at mesoderm invagination. First, bypassing the need for force-mediated activation by ectopically expressing NICD in embryos depleted of full length Notch did not prevent the gastrulation-dependent transition in levels. Indeed, it appears that the elevated transcription at gastrulation may be able to overcome any intrinsic attenuation mechanism that switches off NICD induced transcription (*Viswanathan et al., 2021*). Second, an unrelated enhancer, *vnd*, also exhibited an increase in levels at the time of gastrulation that was correspondingly perturbed in *fog* mutants or α-Cat depleted embryos. Altogether, our results argue in favor of a model where the cell movements associated with gastrulation have a direct impact on transcription in the nucleus, rather than altering the levels of Notch activation per se.

A common feature of both *m5/m8* and *vnd* enhancers is that their activity in MSE cells is more profoundly influenced by gastrulation than elsewhere in the embryo. This could be due to differences in the context of transcription factors and other regulators present in the MSE cells, that make them more sensitive, or to variations in the magnitude of the mechanical forces exerted by gastrulation. In favor of the latter, ectoderm cells are thought to have different mechanical properties to the mesoderm

and the lateral cells, which contribute to the normal progression of gastrulation (*Rauzi et al., 2015*). We therefore hypothesize that transcription levels can be modulated by the forces exerted on cells, and that this effect may be specific to certain classes of enhancers. Here we have found that although both *m5/m8* and *vnd* enhancers show characteristics of such regulation, the Notch-dependent *m5/m8* appears the more sensitive. We propose that mechanisms acting at the level of the nucleus will be responsible for transmitting this force-mediated regulation.

To exert effects on nuclear functions, the mechanical changes induced by gastrulation need to be transmitted to the nucleus. The mechanotransduction cascade is thought to involve the cytoskeleton and the LINC complex (*Chang et al., 2015*). Along with myosin, the LINC complex is proposed to transmit information about substrate stiffness to the transcriptional machinery (*Alam et al., 2016*). A similar mechanism could be involved in the gastrulating embryo to transduce mechanical information at the cell level to changes in transcription, based on the fact that coupling between transcription and gastrulation was no longer evident when the LINC complex was depleted. Although the fact that there are also changes in gastrulation under these conditions make it difficult to conclude that there is a direct effect of the LINC complex, the results are nevertheless consistent with the model. We explored two downstream mechanisms that could ultimately be responsible for increased transcriptional activity. The first was a change in nuclear import, either a general increase in permeability of nuclear factors or a more specific effect on key transcription factors such as Twist. No change in general nuclear import properties were detected and, although a gradual increase in Twist was detected, there was no step change at gastrulation that could account for the sharp increase in transcription. The second was a change in chromatin accessibility. We used some simple methods to probe chromatin compaction and mobility, which indicated that the chromatin becomes more mobile at gastrulation. Higher resolution approaches at the single cell level would be needed to ascertain whether these changes are indeed influenced by gastrulation. Finally, it is also possible that subtle changes in several different of these processes are required to bring about the step-change in transcription, and that each alone makes only a minor contribution.

There is increasing evidence that substrate stiffness and mechanical forces can influence cell fate decisions (*Alam et al., 2016*; *Wei et al., 2015*; *Roy et al., 2018*; *Muncie et al., 2020*). Our data showing a connection between gastrulation and transcription levels support the suggestion that there are mechanisms connecting the mechanical properties experienced by cells with the transcriptional output. In the mesectoderm, these effects may be important to ensure that a robust and straight boundary is positioned at the midline. Such a connectivity, between forces and transcription levels, is likely to be of major significance in building tissues and organs, to ensure that the correct structure and shape are adopted in the context of the developing organism.

## Materials and methods
### Fly strains and genetics
The MS2 reporter *m5/m8* [II] (*m5/m8-peve-24xMS2-lacZ-SV40[25C]*), inserted in the landing site attP40-25C in the second chromosome (*Bischof et al., 2013*), other reporters used in (*Figure 1*) and the *eve2-FRT-STOP-FRT-NICD*[51D] construct used to ectopically express NICD have been previously described (*Falo-Sanjuan et al., 2019*). *vnd-MS2* (*vndEEE-peve-24xMS2-lacZ-SV40*) was generated by replacing *m5/m8* for the *vndEEE* enhancer, as defined by *Stathopoulos et al., 2002*, using primers GGGAAGCTTGGGTAAGCACAAGGATTCC and GGGACCGGTCGAATAAGCTGCAAGGAGATC with *Hind*III and *Age*I sites. The resulting plasmid was inserted in the same attP landing site (attP40-25C) as *m5/m8* by ΦC31 mediated recombination (*Bischof et al., 2013*). Full genotypes of used lines are detailed in *Table 1*.

### *fog* mutant background
To test expression from *m5/m8* and *vnd* in the *fog* mutant background, third chromosome recombinants *His2av-RFP, nos-MCP-GFP* (*Falo-Sanjuan et al., 2019*) were combined with a *fog* null allele (fog[S4], BDSC #2100). Control embryos were obtained by crossing *His2av-RFP, nos-MCP-GFP* females with *m5/m8* or *vnd* males. Hemizygous *fog* mutant embryos were obtained by crossing *fog[S4] / FM6;; His2av-RFP, nos-MCP-GFP* with *m5/m8* or *vnd* males and they were recognized by the presence of ectopic folds after gastrulation and embryonic lethality. Other embryos obtained from the same

**Table 1.** Full genotypes and sources of used *Drosophila* lines.

| Name (Chr) | Full genotype | Source |
|---|---|---|
| *His2Av::RFP (III)* | w[*];P{w[+mC]=His2Av-mRFP1}III.1 | BDSC #23650 |
| *His2Av::RFP; nos-MCP::GFP* | y[1] w[*]; P{w[+mC]=His2Av-mRFP1}II.2; P{w[+mC]=nos MCP.EGFP}2 | BDSC #60340 |
| *His2Av::iRFP, nos-MCP::mCherry (II)* | yw; nos-MCP-NoNLS-mCherry, Histone-iRFP | *Eck et al., 2020*; *Liu et al., 2021* |
| *nos-MCP::mCherry (III)* | yw;; nos-MCP-NoNLS-mCherry | *Liu et al., 2021* |
| *vasa-mCherry (III)* | yw; pCasper-vasa-mCherry | YJ. Kim, J. Zhao, HG. Garcia, unpublished results |
| *Gap43::mCherry (I)* | Pw[+mC]=sqhp-Gap43::mCherry | *Izquierdo et al., 2018* |
| *Nup107::RFP (III)* | w[*]; wg[Sp-1]/CyO; P{w[+mC]=mRFP-Nup107.K}7.1 | BDSC #35517 |
| *αTub-Gal4::VP16 (II)* | w[*]; P{w[+mC]=matalpha4-GAL-VP16}V2H | BDSC #7062 |
| *ovo-FLP (I)* | P{w[+mC]=ovo FLP.R}M1A, w[*] | BDSC #8727 |
| *betaTub85D-FLP (I)* | P{ry[+t7.2]=betaTub85D-FLP}1; ry[506] | BDSC #7196 |
| *m5/m8-MS2 (II)* | w; P{w[+mC]=m5/m8-peve-24xMS2-lacZ-SV40}attP40 | *Falo-Sanjuan et al., 2019* |
| *vnd-MS2 (II)* | w; P{w[+mC]=vndEEE-peve-24xMS2-lacZ-SV40}attP40 | This study |
| *eve2-NICD (II)* | w; P{w[+mC]=2xeve2*-FRT-STOP-FRT-NICD-eve3'UTR}attP51D | *Falo-Sanjuan et al., 2019* |
| *Twi-RedLlamaTag (II)* | y[1] w[1,118]; Twi-JB10 /CyO; | *Bothma et al., 2018* |
| *fog[S4] (I)* | y[1]() fog[S4]/FM7c | BDSC #2100 |
| *w RNAi Valium22 (III)* | y[1] sc[*] v[1]; P{y[+t7.7] v[+t1.8]=TRiP .GL00094}attP2 | BDSC #35573 |
| *α-Cat RNAi Valium20 (III)* | y[1] sc[*] v[1] sev[21](); P{y[+t7.7] v[+t1.8]=TRiP.HMS00317}attP2 | BDSC #33430 |
| *cta RNAi Valium20 (II)* | y[1]() sc[*] v[1]; P{y[+t7.7] v[+t1.8]=TRiP.HMC03421}attP40 | BDSC #51848 |
| *RhoGEF2 RNAi Valium20 (III)* | y[1] sc[*] v[1]; P{y[+t7.7] v[+t1.8]=TRiP.HMS01118}attP2 | BDSC #34643 |
| *Notch RNAi Valium20 (III)* | y[1] v[1]; P{y[+t7.7] v[+t1.8]=TRiP.HMS00009}attP2 | BDSC #33616 |
| *zld RNAi Valium20 (II)* | y[1] sc[*] v[1]; P{y[+t7.7] v[+t1.8]=TRiP.HMS02441}attP40 | BDSC #42016 |
| *grh RNAi Valium22 (III)* | y[1] sc[*] v[1]; P{y[+t7.7] v[+t1.8]=TRiP .GL01069}attP2 | BDSC #36890 |
| *klar RNAi Valium20 (III)* | y[1] sc[*] v[1] sev[21]; P{y[+t7.7] v[+t1.8]=TRiP.HMS01612}attP2 | BDSC #36721 |
| *koi RNAi Valium20 (II)* | y[1] sc[*] v[1] sev[21]; P{y[+t7.7] v[+t1.8]=TRiP.HMS02172}attP40 | BDSC #40924 |
| *lamC RNAi Valium20 (II)* | y[1] sc[*] v[1] sev[21]; P{y[+t7.7] v[+t1.8]=TRiP.HMC04816}attP40 | BDSC #57501 |
| *zld::GFP (III)* | w[1,118]; PBac{y[+mDint2] w[+mC]=zld GFP.FPTB}VK00033 /TM3, Sb[1] | BDSC #51350 |
| *Su(H)::GFP (III)* | M{Su(H).WT.EGFP}attP86F | *Gomez-Lamarca et al., 2018* |
| *H::GFP (II)* | M{H.WT.EGFP}51D | *Gomez-Lamarca et al., 2018* |
| *nej::venus* | w[1,118] PBac{602 .P.SVS-1}nejCPTI000727 | Kyoto #115119 |

*Table 1 continued on next page*

*Table 1 continued*

| Name (Chr) | Full genotype | Source |
|---|---|---|
| *Med4::GFP* | PBac{fTRG00975.sfGFP-TVPTBF}VK00002 | VDRC #v318702 |
| *Med17:GFP* | PBac{fTRG00979.sfGFP-TVPTBF}VK00002 | VDRC #v318704 |
| *skd::venus* | w[1,118]; PBac{681 .P.FSVS-1}skdCPTI001170 | Kyoto #115130 |
| *vasa-eGFP (III)* | yw; pCasper-vasa-eGFP | *Kim et al., 2021* |
| *hs-FLP;; ubi-nls::GFP, FRT80E* | Pry[+t7.2]=hsFLP22, y[1] w[*]; Kr[If-1]/CyO; Pw[+mC]=Ubi-GFP(S65T) nls3L PBacw[+mC]=WHND-MLRQ[f00651] | BDSC #76329 |
| *nos-nls::PCP::GFP* | | Courtesy from Tim Weil and Liz Gavis |

cross were grouped together and labeled *fog*[+/+/-]. To combine the *fog* background with ectopic NICD produced by *eve2-NICD*, *fog[S4]* was combined with *eve2-FRT-STOP-FRT-NICD* and crossed with *ovo-FLP;; His2av-RFP, nos-MCP-GFP*. F1 females containing all transgenes were crossed with *m5/m8-MS2* males; and induce removal of the FRT-STOP-FRT cassette in the germline so that NICD is produced in the *eve2* pattern in F2 embryos. Controls for this experiment were obtained by crossing *eve2-FRT-STOP-FRT-NICD* with *ovo-FLP;; His2av-RFP, nos-MCP-GFP* and F1 females with *m5/m8-MS2* males.

## Maternal KD

The maternal driver *αTub-Gal4::VP16* (BDSC # 7062) was combined with *His2av-RFP, nos-MCP-GFP* to generate *αTub-Gal4::VP16; His2Av-RFP, nos-MCP-GFP*. To knock down genes from the maternal germline, this was crossed with with *UASp-RNAi* lines detailed in *Table 1* or *UASp-w-RNAi* as control (BDSC #35573). Females *αTub-Gal4::VP16 / +; His2Av-RFP, nos-MCP-GFP / UASp-RNAi* or *αTub-Gal4::VP16 /UASp RNAi; His2Av-RFP, nos-MCP-GFP /+*were crossed with *m5/m8-MS2* or *vnd-MS2* to obtain the experimental embryos. To combine this with ectopic NICD, *m5/m8, eve2-FRT-STOP-FRT-NICD* recombinants (*Falo-Sanjuan et al., 2019*) had been previously crossed with *βTub-FLP* and males *βTub-FLP / Y; m5/m8,eve2-FRT-STOP-FRT-NICD / +*, which induce FRT-STOP-FRT removal in the germline, crossed with females in which germline KD occurs. In this way, the resulting embryos express both MS2 and *eve2-NICD*, and candidate genes have been maternally knocked down. To quantify the degree of maternal Klar and Koi KD, *αTub-Gal4::VP16* was crossed with the same lines and F2 embryos were collected for RT-qPCR. Levels of Zld KD were checked using a Zld-GFP line. Other RNAi lines have been previously used (*Falo-Sanjuan and Bray, 2021*; *Garcia De Las Bayonas et al., 2019*; *Wingen et al., 2017*; *Zraly et al., 2020*) or produced a very strong phenotype. Crosses performed for each experiment are detailed in *Table 2*.

## Live imaging

Embryos were collected on apple juice agar plates with yeast paste, dechorionated in bleach and mounted in Voltalef medium (Samaro) between a semi-permeable membrane and a coverslip. The ventral side of the embryo was facing the coverslip. Videos were acquired in a Leica SP8 confocal using a 40 x apochromatic 1.3 objective, zoom x2 and 400 × 400 px size (providing an XY resolution of 0.36 µm/px), 12 bit depth, 400 Hz image acquisition frequency and pinhole of 4 airy units. 27, 29 or 32 × 1 µm slices were collected, with total acquisition time of 15–20 s per frame, depending on the experiment. Different genetic conditions are only compared in the same plot if the same microscope and settings were used. To image embryos in which nuclear levels of different factors were going to be quantified, multiple embryos were acquired at once using the multi-position setting in the microscope and a final time resolution of 1 or 1.5 min, which also helped reduce bleaching. Depending on the condition only GFP or GFP and His2Av-RFP were imaged.

## Dextrans injection

Embryos obtained from crossing *His2Av-RFP /+* females and males were dechorionated, glued with heptane glue to a coverslip, and covered with Voltalef medium (Samaro). Injections were performed only on pre-nc10 embryos using pulled glass needles and dextrans diluted in injection buffer (0.1 mM

**Table 2.** Crosses to obtain embryos used in each experiment.

| Cross | Figure |
|---|---|
| ♀ His2Av::RFP, nos-MCP::GFP × ♂ m5/m8-peve-MS2-lacZ-SV40[attP40,II] **Falo-Sanjuan et al., 2019** | (Figure 1, Figure 1—figure supplement 1) |
| ♀ His2Av::RFP, nos-MCP::GFP × ♂ m5/m8-peve-MS2-lacZ-SV40[attP2,III] **Falo-Sanjuan et al., 2019** | Figure 1—figure supplement 1 |
| ♀ His2Av::RFP, nos-MCP::GFP × ♂ sim-peve-MS2-lacZ-SV40[attP40,II] **Falo-Sanjuan et al., 2019** | Figure 1—figure supplement 1 |
| ♀ His2Av::RFP, nos-MCP::GFP × ♂ m5/m8-pm7-MS2-lacZ-SV40[attP40,II] **Falo-Sanjuan et al., 2019** | Figure 1—figure supplement 1 |
| ♀ His2Av::RFP, nos-MCP::GFP × ♂ m5/m8-psimE-MS2-lacZ-SV40[attP40,II] **Falo-Sanjuan et al., 2019** | Figure 1—figure supplement 1 |
| ♀ αTub-Gal4::VP16 / +; His2Av::RFP, nos-MCP::GFP / UASp-w RNAi ♂ m5/m8-peve-MS2-lacZ-SV40[attP40,II] | Figures 2; 5; 6, Figure 2—figure supplements 1; 2; 4, Figure 3—figure supplement 1, Figure 5—figure supplements 2, Figure 6—figure supplement 3 |
| ♀ αTub-Gal4::VP16 / +; His2Av::RFP, nos-MCP::GFP / UASp-α-Cat RNAi ♂ m5/m8-peve-MS2-lacZ-SV40[attP40,II] | Figures 2; 6, Figure 2—figure supplements 1; 2 |
| ♀ His2Av::RFP, nos-MCP::GFP (III) × ♂ m5/m8-peve-MS2-lacZ-SV40[attP40,II] | Figure 2, Figure 2—figure supplements 1; 2, Figure 6—figure supplement 3 |
| ♀ fog[S4] / FM6;, His2Av::RFP, nos-MCP::GFP × ♂ m5/m8-peve-MS2-lacZ-SV40[attP40,II] | Figure 2, Figure 2—figure supplements 1; 2, Figure 6—figure supplement 3 |
| ♀ αTub-Gal4::VP16 / UASpcta RNAi; His2Av::RFP, nos-MCP::GFP / + ♂ m5/m8-peve-MS2-lacZ-SV40[attP2,III] | Figure 2, (Figure 2—figure supplements 1; 2) , Figure 6—figure supplement 3 |
| ♀ αTub-Gal4::VP16 / +; His2Av::RFP, nos-MCP::GFP / UASp-RhoGEF2 RNAi ♂ m5/m8-peve-MS2-lacZ-SV40[attP40,II] | Figure 2, (Figure 2—figure supplements 1; 2) , Figure 6—figure supplement 3 |
| ♀ αTub-Gal4::VP16 / +; His2Av::RFP, nos-MCP::GFP / UASp-w RNAi ♂ m5/m8-peve-MS2-lacZ-SV40[attP2,III] **Falo-Sanjuan and Bray, 2021** | Figure 2—figure supplement 3 |
| ♀ αTub-Gal4::VP16 / +; His2Av::RFP, nos-MCP::GFP / UASp-α-Cat RNAi ♂ m5/m8-peve-MS2-lacZ-SV40[attP2,III] **Falo-Sanjuan and Bray, 2021** | Figure 2—figure supplement 3 |
| ♀ αTub-Gal4::VP16 / +; His2Av::RFP, nos-MCP::GFP / UASp-w RNAi ♂ sim-peve-MS2-lacZ-SV40[attP40,II] | Figure 2—figure supplement 3 |
| ♀ αTub-Gal4::VP16 / +; His2Av::RFP, nos-MCP::GFP / UASp-α-Cat RNAi ♂ sim-peve-MS2-lacZ-SV40[attP40,II] | Figure 2—figure supplement 3 |
| ♀ αTub-Gal4::VP16 / UASpzld RNAi; His2Av::RFP, nos-MCP::GFP / + ♂ m5/m8-peve-MS2-lacZ-SV40[attP40,II] | Figure 2—figure supplement 4 |
| ♀ αTub-Gal4::VP16 / +; His2Av::RFP, nos-MCP::GFP / UASp-grh RNAi ♂ m5/m8-peve-MS2-lacZ-SV40[attP40,II] | Figure 2—figure supplement 4 |
| ♀ × ♂ αTub-Gal4::VP16 / +; zld::GFP / UASp-w RNAi | Figure 2—figure supplement 4 |
| ♀ × ♂ αTub-Gal4::VP16 / UASpzld RNAi; zld::GFP / + | Figure 2—figure supplement 4 |
| ♀ His2Av::RFP, nos-MCP::GFP (III) × ♂ vndEEE-peve-MS2-lacZ-SV40[attP40,II] | Figure 3, Figure 3—figure supplement 1 |
| ♀ αTub-Gal4::VP16 /+; His2Av::RFP, nos-MCP::GFP / UASp-w RNAi ♂ vndEEE-peve-MS2-lacZ-SV40[attP40,II] | Figure 3, Figure 3—figure supplement 1 |
| ♀ αTub-Gal4::VP16 / +; His2Av::RFP, nos-MCP::GFP / UASp-α-Cat RNAi ♂ vndEEE-peve-MS2-lacZ-SV40[attP40,II] | Figure 3, Figure 3—figure supplement 1 |
| ♀ αTub-Gal4::VP16 / +; His2Av::RFP, nos-MCP::GFP / UASp-Notch RNAi ♂ m5/m8-peve-MS2-lacZ-SV40[attP40,II] | Figure 3—figure supplement 1 |

*Table 2 continued on next page*

Table 2 continued

| Cross | Figure |
|---|---|
| ♀ αTub-Gal4::VP16 / +; +; His2Av::RFP, nos-MCP::GFP / UASp-Notch RNAi ♂ vndEEE-peve-MS2-lacZ-SV40[attP40,II] | *Figure 3, Figure 3—figure supplement 1* |
| ♀ fog[S4] / FM6;; His2Av::RFP, nos-MCP::GFP x ♂ vndEEE-peve-MS2-lacZ-SV40[attP40,II] | *Figure 3, Figure 3—figure supplement 1* |
| ♀ ovo-FLP / +; eve2-FRT-STOP-FRT-NICD / +; His2Av::RFP, nos-MCP::GFP / + ♂ m5/m8-peve-MS2-lacZ-SV40[attP40,II] | *Figure 4, Figure 4—figure supplement 1* |
| ♀ ovo-FLP / fog[S4]; eve2-FRT-STOP-FRT-NICD / +; His2Av::RFP, nos-MCP::GFP / + ♂ m5/m8-peve-MS2-lacZ-SV40[attP40,II] | *Figure 4, Figure 4—figure supplement 1* |
| ♀ αTub-Gal4::VP16 / +; +; His2Av::RFP, nos-MCP::GFP / UASp-w RNAi ♂ βTub85D-FLP; m5/m8-peve-MS2-lacZ-SV40, eve2-FRT-STOP-FRT-NICD / + | *Figure 4, Figure 4—figure supplement 1* |
| ♀ αTub-Gal4::VP16 / +; +; His2Av::RFP, nos-MCP::GFP / UASp-Notch RNAi ♂ βTub85D-FLP; m5/m8-peve-MS2-lacZ-SV40, eve2-FRT-STOP-FRT-NICD / + | *Figure 4, Figure 4—figure supplement 1* |
| ♀ αTub-Gal4::VP16 / +; +; His2Av::RFP, nos-MCP::GFP / UASp-α-Cat RNAi ♂ βTub85D-FLP; m5/m8-peve-MS2-lacZ-SV40, eve2-FRT-STOP-FRT-NICD / + | *Figure 4, Figure 4—figure supplement 1* |
| ♀ αTub-Gal4::VP16 / +; +; His2Av::RFP, nos-MCP::GFP / UASp-klar RNAi ♂ m5/m8-peve-MS2-lacZ-SV40[attP40,II] | *Figure 5* |
| ♀ x ♂ αTub-Gal4::VP16 / +; UASp-w RNAi / + (for qPCR) | *Figure 5—figure supplement 2* |
| ♀ x ♂ αTub-Gal4::VP16 / UASpkoi RNAi; +/+ (for qPCR) | *Figure 5—figure supplement 2* |
| ♀ x ♂ αTub-Gal4::VP16 / +; UASp-klar RNAi / + (for qPCR) | *Figure 5—figure supplement 2* |
| ♀ Gap43::mCherry; nos-MCP::GFP x ♂ m5/m8-peve-MS2-lacZ-SV40[attP40,II] *Falo-Sanjuan and Bray, 2021* | *Figure 5—figure supplement 1* |
| ♀ x ♂ His2Av::iRFP, nos-MCP::mCherry / CyO; nos-MCP::mCherry, vasa-eGFP | *Figure 6—figure supplement 1* |
| ♀ x ♂ ubi-nls::GFP, FRT80E | *Figure 6—figure supplement 1* |
| ♀ x ♂ nos-nls::PCP::GFP | *Figure 6—figure supplement 1* |
| ♀ x ♂ His2Av::RFP / + (dextran injected) | *Figure 6—figure supplement 1* |
| ♀ x ♂ med4-GFP | *Figure 6—figure supplement 1* |
| ♀ x ♂ med30-GFP | *Figure 6—figure supplement 1* |
| ♀ x ♂ skd-venus | *Figure 6—figure supplement 1* |
| ♀ His2Av::iRFP, nos-MCP::mCherry / CyO; nos-MCP::mCherry, Su(H)::GFP ♂ m5/m8-peve-MS2-lacZ-SV40[attP40,II] | *Figure 6—figure supplement 1* |
| ♀ x ♂ H::GFP; His2Av::RFP | *Figure 6—figure supplement 1* |
| ♀ x ♂ nej-venus | *Figure 6—figure supplement 1* |
| ♀ Vasa-mCherry (III) x ♂ Twi-JB10/CyO | *Figure 6* |
| ♀ Nup107::RFP, nos-MCP::GFP x ♂ m5/m8-peve-MS2-lacZ-SV40[attP40,II] | *Figure 6, Figure 6—figure supplement 2* |

**Table 3.** Primers used for qPCR.

| Primer | Sequence |
| --- | --- |
| *klar FWD 1* | GTCTTGCCAAGACATGGATG |
| *klar REV 1* | GGCTGGTCGACTGAATCTTG |
| *koi FWD 1* | AGCTGGAGACCACACAAAAC |
| *koi REV 1* | CGTCTTGGGAGTTTTGTTCC |
| *koi FWD 2* | GGAACAAAACTCCCAAGACG |
| *koi REV 2* | TCTGCTGGACCATGTAGTTG |
| *RpII215 FWD* | GACTCGACTGGAATTGCACC |
| *RpII215 FWD* | TCTTCATCGGGATACTCGCC |

sodium phosphate buffer, 5 mM KCl, *Sharrock et al., 2021*) at a concentration of 0.125 mg/ml for Dextran-40kDa-FITC (Sigma-Aldrich FD40S) and 0.25 mg/ml for Dextran-70kDa-FITC (Sigma-Aldrich 46945).

## mRNA extraction and qPCR

RT-qPCR quantification of maternal KD was performed as previously described (*Falo-Sanjuan and Bray, 2021*). Embryos were dechorionated in bleach and early embryos (pre-nc10) / eggs were selected in Voltalef medium. Pools of 15–20 embryos of each genotype were transferred to eppendorf tubes and dissociated in TRI Reagent (Sigma) with a plastic pestle. mRNA was extracted by adding chloroform, 10 min centrifugation at 4°C and let to precipitate with isopropanol overnight. DNA was then pelleted by 10 min centrifugation at 4°C, washed in 70% ethanol, dried and resuspended in DEPC-treated water. Approximately 2 μg of RNA from each sample were DNAse treated with the DNA-free DNA Removal Kit (Invitrogen) in the presence of RiboLock RNase Inhibitor (Thermo Scientific). 1 μg of DNA-free RNA was then used for reverse transcription using M-MLV Reverse Transcriptase (Promega) in the presence of RiboLock. RT-qPCR was performed using SYBR Green Mastermix (Sigma) and primers detailed in *Table 3*.

### Image analysis

#### Tracking nuclei and MS2 quantification

Videos were analyzed using custom MATLAB (MATLAB R2020a, MathWorks) scripts that have been previously described (*Falo-Sanjuan et al., 2019*; *Falo-Sanjuan and Bray, 2021*). Briefly, the His2Av-RFP signal was used to segment and track the nuclei in 3D. Each 3D stack was first filtered using a 3D median filter of 3 and increasing the contrast based on the intensity profile of each frame to account for bleaching. This was followed by Fourier transform log filter to enhance round objects (*Garcia et al., 2013*), and segmentation by applying a fixed intensity threshold, which was empirically determined. Filters to fill holes in objects, exclusion based on size and 3D watershed accounting for anisotropic voxel sizes (*Mishchenko, 2013*) were used to remove miss-segmented nuclei and split merged nuclei. The final segmented stack was obtained by filtering by size again and thickening each object. Lastly, each nuclei in the segmented stack was labeled and the position of each centroid (in X, Y and Z) was calculated for tracking. Nuclei were tracked in 3D by finding the nearest object (minimum Euclidean distance between two centroids in space) in the previous 2 frames which was closer than 6 μm. If no object was found, that nucleus was kept with a new label. If more than one object was 'matched' with the same one in one of the previous 2 frames, both were kept with new labels. After tracking, the positions of all pixels from each nucleus in each frame were used to measure the maximum fluorescence value in the GFP channel, which was used as a proxy of the spot fluorescence. Note that when a spot cannot be detected by eye this method detects only background, but the signal:background ratio is high enough that the subsequent analysis allows to classify confidently when the maximum value is really representing a spot.

#### Tracking cell membranes and quantification of cell properties

Videos from embryos expressing the membrane marker *Gap43-mCherry* (*Izquierdo et al., 2018*) combined with the MS2 system to measure *m5/m8 [II]* transcription from *Falo-Sanjuan and Bray, 2021* were reanalyzed to correlate cell properties with transcription. First, videos were locally-projected to corrected for the curvature of the embryo using the Fiji plugin LocalZProjector (*Herbert et al., 2021*). All slices from the MCP-GFP channel were used to obtain a maximum projection whereas 3 medial-apical slices (4 μm) from the *Gap43-mCherry* were used to obtain a maximum intensity projection. Membranes were segmented using TissueAnalyzer (*Aigouy et al., 2016*) and MS2 spots were segmented using the Weka Segmentation Tool (*Arganda-Carreras et al., 2017*), both in Fiji. The following analysis was carried out in Matlab, where cells and spots were independently tracked

and then spots were assigned to cells based on what cell each spot most commonly overlapped with. Cells assigned to MS2 spots were used to define the mesectoderm stripes and cells located in between them were labeled as mesoderm. Mesectoderm cells were then classified as 'active' or 'active +increasing' based on their transcription profiles and the corresponding properties for the cells assigned to them extracted and plotted. Length of contact with ME cells was calculated by adding the number of pixels intersected between each cell thickened by 1 pixel with all the ME cells also thickened by 1 pixel.

## Tracking and quantification of nuclear concentrations

Depending on which factor was being analyzed, some videos expressed only the fluorescently-tagged factor of interest (if the nuclear levels were high enough to use for nuclear segmentation) or in combination with *His2Av-RFP* or *His2Av-iRFP*. In the first case (*nls-GFP, nls-PCP-GFP, Med4-GFP, Med30-GFP*, Skd-venus, Nej-venus and Twist-LlamaTag), nuclei were directly segmented from the channel in which nuclear levels were going to be quantified and the average nuclear intensity quantified over time. In the other cases (GFP, dextran injections, Su(H)-GFP and H-GFP), nuclei were segmented and tracked based on the His2Av signal and the average GFP/FITC intensity quantified in each nucleus. We note that because in most cases all cells in the embryo contained these factors, the signal could not be normalized to another region to account for bleaching, and therefore the fluorescent profiles might have been reduced over time due to bleaching.

## Nuclear membrane segmentation and quantification of morphological parameters

Nuclear membranes tagged with Nup107-RFP (*Katsani et al., 2008*) were segmented in Imaris (Oxford Instruments) using the cell segmentation tool and manual correction. Cells were then classified as ME, MSE or NE by marking the overlap with MS2 signal (MSE) and cells dorsal and ventral to them. Morphological properties of the cell surfaces were extracted from Imaris and plotted in Matlab for each cell population.

## Data processing and statistical analysis

### MS2 data processing

Processing of MS2 data (definition of active nuclei and normalization for bleaching) has been carried out as described in our previous work (*Falo-Sanjuan et al., 2019*). After the fluorescent trace of each nucleus over time was obtained, only nuclei tracked for more than 10 frames were retained. Then, nuclei were classified as inactive or active. To do so, the average of all nuclei (active and inactive) was calculated over time and fitted to a straight line. A median filter of 3 (i.e. over a period of 45–60") was applied to each nucleus over time to smooth the trace and ON periods were considered when fluorescent values were 1.2 times the baseline at each time point. This produced an initial segregation of active (nuclei ON for at least 5 frames, i.e. 1.25–1.6 min) and inactive nuclei. These parameters were determined empirically on the basis that the filters retained nuclei with spots close to background levels and excluded false positives from bright background pixels, which were validated by overlaying nuclei defined by the pipeline as actively transcribing with the MS2 signal, and confirming that the two were concordant. The mean fluorescence from MCP-GFP in the inactive nuclei was then used to define the background baseline and active nuclei were segregated again in the same manner. The final fluorescence values in the active nuclei were calculated by removing the fitted baseline from the maximum intensity value for each, and normalizing for the percentage that the MCP-GFP fluorescence in inactive nuclei decreased over time to account for the loss of fluorescence due to bleaching. Nuclei active in cycles before nc14 were discarded based on the timing of their activation.

In all videos, time into nc14 was considered from the end of the 13th syncytial division. To avoid early stochastic activity, transcription was considered from 15 min into nc14 in most experiments, except for expression in the presence of maternal Gal4, in which it was considered from 30 min to exclude earlier stochastic activity induced by *Gal4VP16*. For *vnd*, transcription was considered all throughout nc14. The total mRNA output (in AU) was obtained by adding all the normalized transcription values for each cell in a defined time period. In plots were cells were classified as 'active' or 'active +increasing', nuclei which increase in levels were defined if the average intensity from 15 min of the start of ME invagination was higher than the average of all active nuclei at that point and at

least 1.7 times higher than the average intensity in the 15 min prior to ME invagination. Using these values, most nuclei were classified in a way that matched what could be observed by looking directly at the transcription traces.

## Quantifying gastrulation progression and changes in transcription levels

Start of apical constriction, start of mesoderm invagination and end of gastrulation were manually defined for each embryo from plots showing the movement of MSE cells. Transcribing nuclei in each region were selected and the average movement of their centroids in the Y axis, corresponding to the DV axis in the embryo, was calculated. This produces a plot with one or two peaks of movement. A large peak of movement is produced during ME invagination, as MSE cells move ventrally. If the embryo is mounted completely ventrally, only this peak of movement is observed. If the orientation of the embryo is tilted, the whole embryo rolls inside the vitelline membrane during apical constriction, which is detected as an earlier peak of movement of cells dorsally. After that, the second peak of movement corresponds with mesoderm invagination. The transition between the two movements, corresponds to and was used to define the start of ME invagination. Similarly, transition point in mean transcription levels was manually defined from each plot showing the mean fluorescence of selected cells, by selecting the inflection point between two different levels of transcription. Only embryos that showed a clear transition in levels were used in the correlation plots, although we note that only in most cases all embryos from each genetic condition were included.

## Quantifying histone intensity and MS2 spot movement

To quantify average histone intensity around the MS2 spot, spots were segmented using the Differential-of-Gaussians approach previously described (*Garcia et al., 2013*). A 5 × 5 pixel window surrounding the MS2 spot was then extracted from the *His2Av-RFP* channel and the average intensity calculated.

## Statistical analysis

In figures and figure legends, n number indicates number of embryos imaged for each biological condition. Where appropriate, n number next to heatmaps indicates total number of cells combining all embryos for each biological condition. Plots showing mean levels of transcription and SEM (standard error of the mean) combine all traces from multiple embryos from the same biological condition.

### Reagents and software availability

The MATLAB app to track nuclei, quantify MS2 traces and define properties of gastrulation can be obtained at https://github.com/BrayLab/LiveTrx (*Falo-Sanjuan, 2021*; copy archived at swh:1:rev:564d8ac59d4dcf2cd5167960e5ae97d4e8932647).

### Acknowledgements

We thank members of the Bray Lab for helpful discussions. Thanks to Yang Joon Kim, Jiaxi (Jake) Zhao for sharing unpublished flies, Tom Sharrock for advice on embryo injections, members of the Sanson, Weil and Garcia labs for providing flies and advice and to Kat Millen and the Genetics Fly Facility for injections. We acknowledge the Cambridge Advanced Imaging Centre for their support, assistance in this work and use of their microscopes. This work was supported by a Wellcome Trust Investigator Award (212207/Z/18/Z) and a Medical Research Council Programme grant (MR/T014156/1) and by a PhD studentship to JF-S from the Wellcome Trust (109144/Z/15/Z).

## Additional information

### Funding

| Funder | Grant reference number | Author |
| --- | --- | --- |
| Wellcome Trust | 212207/Z/18/Z | Sarah Bray |

| Funder | Grant reference number | Author |
|---|---|---|
| Medical Research Council | MR/T014156/1 | Sarah Bray |
| Wellcome Trust | 109144/Z/15/Z | Julia Falo-Sanjuan |

The funders had no role in study design, data collection and interpretation, or the decision to submit the work for publication. For the purpose of Open Access, the authors have applied a CC BY public copyright license to any Author Accepted Manuscript version arising from this submission.

### Author contributions
Julia Falo-Sanjuan, Conceptualization, Investigation, Methodology, Software, Validation, Visualization, Writing – original draft, Writing – review and editing; Sarah Bray, Conceptualization, Funding acquisition, Project administration, Supervision, Visualization, Writing – original draft, Writing – review and editing

### Author ORCIDs
Julia Falo-Sanjuan http://orcid.org/0000-0002-3563-4789
Sarah Bray http://orcid.org/0000-0002-1642-599X

### Decision letter and Author response
Decision letter https://doi.org/10.7554/eLife.73656.sa1
Author response https://doi.org/10.7554/eLife.73656.sa2

## Additional files

### Supplementary files
• Transparent reporting form

• Source data 1. Numerical data plotted in all figures shown in the paper (fluorescence plots, heatmaps, boxplots and correlation plots). Figure, genotypes and axis indicated in each text file.

### Data availability
Source data files are provided for each plot in each figure. Code used for data analysis is available on GitHub: https://github.com/BrayLab/LiveTrx (copy archived at swh:1:rev:564d8ac59d4dcf2c-d5167960e5ae97d4e8932647). Raw movies and analysis files have been deposited in two FigShare repositories: https://doi.org/10.6084/m9.figshare.16619773.v45, https://doi.org/10.6084/m9.figshare.19697413.v2. Datasets Generated: MS2 data from "Notch-dependent and -independent transcription are modulated by tissue movements at gastrulation": Falo Sanjuan J, Bray, SJ, 2022, https://doi.org/10.6084/m9.figshare.16619773.v45, figshare, https://doi.org/10.6084/m9.figshare.16619773.v45. Other data from "Notch-dependent and -independent transcription are modulated by tissue movements at gastrulation": Falo Sanjuan , Bray S, 2022, https://doi.org/10.6084/m9.figshare.19697413.v2, figshare, https://doi.org/10.6084/m9.figshare.19697413.v2.

The following datasets were generated:

| Author(s) | Year | Dataset title | Dataset URL | Database and Identifier |
|---|---|---|---|---|
| Falo-Sanjuan J, Bray SJ | 2022 | MS2 data from "Notch-dependent and -independent transcription are modulated by tissue movements at gastrulation" | https://doi.org/10.6084/m9.figshare.16619773.v45 | figshare, 10.6084/m9.figshare.16619773.v45 |
| Falo-Sanjuan J, Bray SJ | 2022 | Other data from "Notch-dependent and -independent transcription are modulated by tissue movements at gastrulation" | https://doi.org/10.6084/m9.figshare.19697413.v2 | figshare, 10.6084/m9.figshare.19697413.v2 |

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
