## [Editor Report]

In this manuscript, Falo-Sanjuan and Bray provide an elegant set of experiments investigating how cell movements modulate Notch transcriptional activity during *Drosophila* gastrulation. Through a detailed and convincing analysis of live transcriptional reporters and gastrulation-defective mutants, they report a clear example of tissue movements affecting enhancer activity. The idea that morphogenetic movements can regulate the genome through mechanical changes in the nucleus is intriguing and important and is of broad interest to cell and developmental biologists.

---

## [Decision Letter]

**Decision letter after peer review:**

[Editors’ note: the authors submitted for reconsideration following the decision after peer review. What follows is the decision letter after the first round of review.]

Thank you for submitting the paper "Levels of Notch-regulated transcription are modulated by tissue movements at gastrulation" for consideration by *eLife*. Your article has been reviewed by 3 peer reviewers, one of whom is a member of our Board of Reviewing Editors, and the evaluation has been overseen by a Senior Editor. The reviewers have opted to remain anonymous.

Comments to the Authors:

We are sorry to say that, after consultation with the reviewers, we have decided that this work in its current form will not be considered further for publication by *eLife*.

As you will see from their comments, all three reviewers found the subject matter interesting and timely, and considered the data documenting enhancer dynamics to be very high quality. However, the reviewers felt that the observed changes in enhancer activity correlated with cell movements were, on their own, insufficient to warrant publication in *eLife*. Further experimental data connecting enhancer upregulation to mechanical changes on the nucleus are needed. Currently the role of LINC complex, specifically Klar in regulating tension-dependent transcriptional activity at gastrulation, is unclear and the data are at odds with the predictions. Thus, the role of nuclear deformation and Klar’s contribution remain difficult to understand in relation to the proposed model.

Thus, the reviewers felt that the work has potential, but would need extensive additional work. Please note that *eLife* aims to publish articles with a single round of revision that can be accomplished within two months. If you feel that all of the reviewer’s concerns can be addressed in a timely manner we would be open to reviewing a revised submission, but we do not want to delay publication elsewhere, if that is the preferred course of action.

*Reviewer #1:*

This paper describes the intriguing observation that transcriptional activity of a Notch dependent enhancer depends on gastrulation movements in the *Drosophila* embryo. Using the MS2 system to report on live transcriptional dynamics, it is convincingly shown that a burst of transcriptional activity from the Notch-dependent enhancer in cells neighboring the ventral furrow coincides temporally with gastrulation movements, and that such movements are necessary for the transcriptional burst. This gastrulation-dependent upregulation is also observed by NICD-induced transcription, suggesting the mechanism lies downstream of Notch receptor, likely in the nucleus. The idea that morphogenetic movements can affect genome regulation through mechanical changes on the nucleus is intriguing and important, and this paper represents a clear example in the context of embryonic development.

Overall, the data are high quality and are presented in a clear and convincing manner. The data for the most part support the authors conclusions. Specifically, the conclusion that gastrulation movements are required for transcriptional upregulation of Notch-dependent enhancers is well supported by the data, and the use of three different genetic alterations to impair gastrulation is a strength. However, to further demonstrate a causal link between tissue mechanics and transcriptional output, it would be beneficial to also show that ectopic tissue folding or stretch is sufficient to generate an increase in enhancer activity. Currently there is only data showing a loss-of-transcriptional upregulation when tissue folding is impaired, and it is possible that some of the mutants used to block gastrulation have broader pleiotropic effects that could impact Notch signaling. Further, data connecting the observed changes in enhancer activity to mechanical changes on the nucleus are not well supported by the current data.

1. To further demonstrate a causal link between tissue mechanics and transcriptional output, it would be beneficial to also show that ectopic tissue folding or stretch is sufficient to generate an increase transcription. Currently there is only data showing a loss-of-transcriptional upregulation when tissue folding is impaired, and it is possible that some of the mutants used to block gastrulation have broader pleiotropic effects that could impact Notch signaling.

2. A discussion of the possible functional consequences of tissue movements on Notch-dependent transcription at this stage of embryogenesis would be helpful for the non-expert.

3. Data connecting the LINC complex to m5/m8 activity are rather weak, and I'm not sure these data in their current form add much to the paper.

*Reviewer #2:*

In the study presented in this manuscript, Falo-Sanjuan and Bray explore the connection between mechanical forces and transcription during gastrulation, a process involving extensive cell movement, rearrangement as well as tension sensing and specification of different cell lineages. The overall hypothesis and ideas are interesting and valid, the experiments are well designed and data are carefully analyzed. The study starts by explaining how the tension between cells can affect Notch signaling in the mesoectoderm through monitoring MS2 transcriptional activity from the m5/m8 enhancer in control and mutant embryos in which mesoderm invagination fails due to defects in cell-cell adhesion (α-catenin) or actomyosin regulation (RhoGEF2, concertina and fog). These results support the view that cortical cytoskeletal tension is causing an increase in Notch-dependent transcription. Then, somewhat abruptly the working model is revised when full-length Notch is shown not to be required for this increase (eve2-NICD in Notch RNAi); they hypothesize that if Notch signaling does not respond to tension at the cell membrane through ligand-receptor binding, the tension-sensing/gene upregulation at mesoderm invagination must relate to LINC complex function at the nuclear membrane. The transition to study of LINC complex function seems to be a bit jarring and needs more supportive evidence to better explain why alternative explanations were disregarded. Lastly, the title may be a bit misleading as the data support the view that "cell movements associated with gastrulation have a direct impact on transcription in the nucleus"… and that Notch-responses are not the only ones that are affected.

The reviewer had specific concerns (detailed below) that relate to (i) quantitative analyses; (ii) there is an apparent contradiction between title and the fact that Notch appears dispensable for the observed phenotype; (iii) klar RNAi gives the opposite phenotype as expected (but somehow still stated as supporting their model) – this merits more explanation or investigation; and (iv) lack of attention is given to other explanations including possible connection between NICD-Twist-mechanotransduction.

This is a very interesting study but the title, flow of the main text and its figures are somewhat misleading/hard to follow. In particular, the transition to study of LINC complex function seems to be a bit jarring and needs more supportive evidence to better explain why alternative explanations are disregarded. Also, the role of LINC complex, specifically klar in regulating tension-dependent transcriptional activity at gastrulation is unclear, more experiments need to be done to show how it interacts with NICD.

1) MS2 spot calling and signal quantification. The reviewer realizes that the MS2 quantification has been published before but for the current focus on levels of expression, the spot analysis should be revisited. Instead of identifying the MS2 spot, the authors appear to be using the maximum intensity within a nucleus. This ignores the shape of a spot and assumes that the maximum intensity sufficiently represents the spot. A better method of spot detection would include either gaussian fitting or thresholding. Since a key point of the paper is to study an increase in signal that is correlated to gastrulation, additional attention to quantification is necessary.

2) Along these lines, the authors need to provide more information about the number of cells scored for graphs that show "fraction of nuclei…".

3) In Figure 2C,E, only embryos that show a clear change in mean levels of transcription are displayed. What is the total number of embryos collected for each condition? How was a "clear change in mean levels" defined?

4) Figure 2, the control profiles of m5/m8 transcription activity in B, D, F appear to be very different, especially for fog mutants, is there a specific reason?

5) The authors used a vnd enhancer to show that mechanical forces have a general effect on transcription at gastrulation. However, there is no further attempt to explore or discuss how vndEEE might be regulated, through another tension-sensing transcription factor or maybe also by changes in nuclear property/LINC complex? Twist seems likely (see pt 8 below).

6) Figure S4, if m5/m8 is Notch-dependent, what causes the increase at time 0 in the mean profile of Notch RNAi?

7) It is interesting that the ectopic expression of eve2-NICD in Notch RNAi background rescues m5/m8 transcription activity within the eve stripe (Figure 4), but how much of this increase relates NICD versus other tension activated factors (e.g., twist)? Is twist upregulated in the domain of eve2-NICD expression in Notch RNAi embryos? How does this study/timing of m5/m8 upregulation at invagination relate to the mechanical signals discussed in Pouille et al., Sci Signal 2009 (which should be cited).

8) The conclusion that LINC complex/Klar plays a role in tension-regulated Notch signaling activity is poorly supported. Figure 5B shows that there is an early increase in m5/m8 activity, which is, as the authors point out, opposite from the expected result. Therefore, it is necessary to better define the relationship between NICD and nuclear properties modified/influenced by LINC. How does loss of function klar (or other LINC protein) impact intracellular Notch signaling and other tension-related processes, including localization of α-catenin at the AJs, RhoGEF2 at the apical membrane, actomyosin pulsing, expression of cta and fog, etc. Depending on whether twist is found to be upregulated by eve2-NICD (see point 7 above), it might also be worth looking to see how twist expression (upregulation by NICD?) is affected by klar.

9) Unlike m5/m8 enhancer which requires Notch signaling input directly to help activate gene expression, Notch input at the sim enhancer acts to 'derepress' expression by eliminating Su(H) repressor input and allowing output to be responsive to Dorsal and Twist (Zinzen et al., Dev Cell 2006; Morel & Schweisguth, Genes Dev 2000). Therefore, the roles of Notch in m5/m8 and sim enhancers differ. In the current study, the focus is m5/m8 with little done to analyze sim enhancer properties. Interestingly, in Morel & Schweisguth, when Su(H) sites are mutated in a sim enhancer (sim.mut), reporter expression remains in N mutant embryos. It is likely that the sim.mut enhancer would also show upregulated gene expression at mesoderm invagination if tested; and if so may be a better test case for tracking the input that relates to mechanotransduction with subsequent mutagenesis. Possibly this input is Twist.

10) Farge 2003 already showed that Twist increases residency in the nucleus downstream of mechanical induction. Relating to comments above (e.g. point 7 and 9), it is likely that the NICD-effect could be indirect, working to upregulate Twist. If so, this study is still interesting, but are the current results novel?

11) The title is confusing as the data support the view that "cell movements associated with gastrulation have a direct impact on transcription in the nucleus"… and that Notch-responses are not the only ones that are affected.*Reviewer #3:*

This manuscript is of broad interest to readers in the fields of Cell and Developmental Biology. The authors provide an elegant set of experimental evidence demonstrating that Notch transcriptional activity is modulated during *Drosophila* mesoderm movements. The data are of very good quality. The authors could boost their conclusions by better replacing them in the context of their previous studies regarding the role of adherens junctions in Notch transcriptional regulation or turnover (Falo-Sanjuan and Bray, Development 2021). The findings on Klar mutant conditions are highly relevant but more difficult to unit with the interplay between Notch transcriptional control and mesoderm invagination.

The major strength of the manuscript is the quantitative and original analyses of Notch (N) transcriptional levels in conjunction with morphogenetic movements in multiple genetic backgrounds. The authors could improve their manuscript by addressing the following points:

1. In Figure S1, the authors have also quantified N activity with two other reporters of N transcriptional activity. Both reporters show an increase at 50 mins as described in the main text, but both reporters also show a decrease at 55 mins. This decrease is not observed for the line (II) used through the manuscript. One could therefore argue that this drop is also a relevant characteristic of Notch activation during mesoderm invagination. This data should therefore be placed in the main figure and the decrease should be mentioned in the main text. In addition, it will be important that the authors performed the some loss of function analyses (for example RhoGEF2, Fog and Notch mutant conditions) using these two other reporters to further validate their findings.

2. Page 7: The following statement is confusing. "We note the mean levels of transcription from this reporter were not significantly reduced prior to gastrulation, unlike those from another reporter insertion and the endogenous genes that showed decreased levels also during cellularization (Falo-Sanjuan and Bray 2021). The difference likely arises because this m5/m8 [II] insertion achieves maximal transcription at lower levels of Notch activity." In the cited previous paper, the authors have shown that a-cat affects Notch transcription during cellularization. It is unclear why a difference in reporter strengths will lead to distinct conclusions. This needs to be further experimentally explored and explained.

3. Page 12: "As predicted, this treatment greatly reduced transcription from m5/m8 [II] (Figure S4AB).". The reduction is 2-fold. How these fold changes compare with the ones observed in RhoGEF2 and Fog mutant conditions? The Notch mutant conditions should be shown in figure 1. Furthermore, one could argue that a large part of the increase in reporter expression does not reflect an increase in Notch activity. Please discuss.

4. Figure 4. It will be important that the authors plot on the same plot with the same units the endogenous Notch activity in both wt and NotchRNAi.

5. As stated by the authors, Notch transcription findings in Klar mutant conditions are distinct from the authors' prediction since transcription increases prior to mesoderm invagination. The role of nucleus deformation and the Klar contribution remain difficult to understand and insert in the authors proposed model.

6. Statistics: Could the authors propose statistical tests to further support their findings regarding temporal changes in reporter levels and differences between distinct mutant backgrounds?

[Editors’ note: further revisions were suggested prior to acceptance, as described below.]

Thank you for resubmitting your work entitled "Notch-dependent and -independent transcription are modulated by tissue movements at gastrulation" for further consideration by *eLife*. Your revised article has been evaluated by Claude Desplan (Senior Editor) and a Reviewing Editor.

The manuscript has been improved but there are some remaining issues that need to be addressed, as outlined below:

Essential revisions:

Overall, the manuscript is much improved and the authors have addressed most of the reviewers' major concerns. There are two remaining issues that require clarification and additional technical information related to imaging conditions and corrections for nuclear depth over the course of gastrulation, and a previously reported increase in transparency of klar embryos.

1) Please provide additional information about the imaging pipeline – specifically whether any corrections are being made for spatially-dependent changes in signal due to movement of cells in the z-direction. See reviewer 2 comments for details.

2) The klar mutant background is associated with a clearing of embryos, at least in older embryos at stage 8 (Supatto et al. Nature Protocols 2009). Please provide information regarding whether there is an increase in transparency of klar embryos at NC14 that may result in the GFP signal appearing brighter that wild type controls.

*Reviewer #2:*

The manuscript is improved. In particular, the data are more clearly presented and the authors have added more experiments, namely testing whether levels of Twist, chromatin import, chromatin compaction and/or accessibility relates to the increase in mesectoderm gene expression identified at the time of mesoderm invagination.

The author's testing of Twist addresses one of the reviewer's major concerns. While it is still possible that post translational modification of Twist acts to increase its activity at invagination, their dismissal of Twist for the purpose of this study is reasonable based on the data presented.

Another major concern was that the use of maximum intensity may lead to noise in the spot calling numbers. The authors state that using the maximum intensity is an accurate method because of tests they have done, however they do not elaborate on what these tests are or how they have verified this beyond the limited use of the alternate method from Garcia 2013. While using the method in Garcia 2013 does show the same general trend, the error is much larger using this method and this doesn't completely support the claim that this method proves the simpler method is correct. The fundamental concern is that single, bright pixels that occur outside of the actual MS2 spot could be skewing the data. An easy way of dealing with this is to apply a median filter, which will remove "salt and pepper" noise to the images. The authors note that they apply this filter across time to smooth the data, however this is would be selecting for nuclei that tend to be on continuously. Another approach would be to manually confirm that the maximum pixel occurs within a spot, but this would be tedious. In the methods section it is stated that "subsequent analysis allows to classify confidently when the maximum value is really representing a spot," however it is unclear how the subsequent analysis alleviates this concern. At minimum, this section of the methods needs to be expanded and made clearer so that the reader can evaluate the method being used.

I do still have a worry that their imaging pipeline is not properly correcting for spatially-dependent changes in signal due to movement of cells in the z-direction. Because they have a H2A-RFP associated with the MCP-GFP MS2 movies, they should use the RFP signal to background correct (not the MCP-GFP signal associated with inactive nuclei). The H2A-RFP signal can vary in space in time due to position of nuclei. The authors are imaging the MS2 signal in a tissue that moves, however they don't provide a correction for this movement or rule out whether this movement is changing intensity of the signal. Specifically, when signal is lower, the MS2 is in a more dorsal location than compared to after invagination where the signal is at the ventral most point. Did the authors correct for any difference in intensity that could be occurring from differences in the position of the tissue? It is conceivable that signal from the ventral most point will be more optically accessible than from points located more dorsally, due to the curvature of the embryo and possibly the diffraction of light as it moves through tissue to reach the objective. An easy way to correct for this would be to use the signal of the H2A signal that was used to segment and track the nuclei, as any positional effects should also affect the H2A signal, and the H2A signal is already mapped to its corresponding spot.

In addition, the klar mutant background is associated with a clearing of embryos such that the GFP signal may appear brighter because embryos are more transparent; this has been shown for older embryos at stage 8 (Supatto et al. Nature Protocols 2009) but it is unclear if this is the case at nuclear cycle 14. If klar mutant embryos at these earlier stages are more transparent, could this account for the apparent increase in GFP signal associated with the klar mutant background (Figure 5B)? Might the H2A-RFP signal serve to normalize the ctr RNAi vs klar RNAi data? Is the H2A-RFP signal also increasing in the klar RNAi? If that is the case, and the klar data is more similar to wildtype – the early phase of transcription may need to be offset vertically (allowing it to align with wildtype). In this alternative scenario, it would appear then that there is a transition late in the klar mutant background. Might the clearing of lipids affect timing of mesoderm invagination and/or when it is recognized/called?

The reviewer appreciates the premise of this study but the authors would be wise to do a bit more controls/investigating to ensure that the observed upregulation in signal associated with mesoderm invagination is not an imaging artifact: (i) might positional effects on MS2 signal be responsible for the increase in GFP observed when cells invaginate/are more directly illuminated; and (ii) to determine whether klar mutants are more transparent and if so use methods to normalize signals obtained in klar in comparison with wildtype.

*Reviewer #3:*

The authors have improved their manuscript and addressed my points. They have added a lot of data related to the dynamics of the nucleus and chromatin mobility. Most of these results are negative but they contribute to the detailed characterization of the process. Overall, the findings support the authors' conclusions.

Nevertheless, the role of Klar remains difficult to fit in the authors' proposed model, since Klar loss of function phenotype is distinct from the ones observed in mutants affecting the mechanical forces associated with gastrulation (α-cat, RhoGEF2 and Fog). I therefore doubt that Klar would be part of the mechanisms by which gastrulation associated forces modulate Notch-dependent and -independent transcriptional regulation.

---

## [Author Response]

[Editors’ note: The authors appealed the original decision. What follows is the authors’ response to the first round of review.]

Reviewer #1:This paper describes the intriguing observation that transcriptional activity of a Notch dependent enhancer depends on gastrulation movements in the *Drosophila* embryo. Using the MS2 system to report on live transcriptional dynamics, it is convincingly shown that a burst of transcriptional activity from the Notch-dependent enhancer in cells neighboring the ventral furrow coincides temporally with gastrulation movements, and that such movements are necessary for the transcriptional burst. This gastrulation-dependent upregulation is also observed by NICD-induced transcription, suggesting the mechanism lies downstream of Notch receptor, likely in the nucleus. The idea that morphogenetic movements can affect genome regulation through mechanical changes on the nucleus is intriguing and important, and this paper represents a clear example in the context of embryonic development.Overall, the data are high quality and are presented in a clear and convincing manner. The data for the most part support the authors conclusions. Specifically, the conclusion that gastrulation movements are required for transcriptional upregulation of Notch-dependent enhancers is well supported by the data, and the use of three different genetic alterations to impair gastrulation is a strength. However, to further demonstrate a causal link between tissue mechanics and transcriptional output, it would be beneficial to also show that ectopic tissue folding or stretch is sufficient to generate an increase in enhancer activity. Currently there is only data showing a loss-of-transcriptional upregulation when tissue folding is impaired, and it is possible that some of the mutants used to block gastrulation have broader pleiotropic effects that could impact Notch signaling. Further, data connecting the observed changes in enhancer activity to mechanical changes on the nucleus are not well supported by the current data.

We are glad that the reviewer considers the results to be intriguing and important and the data to be of high quality. We appreciate the concerns, which we have addressed below.

1. To further demonstrate a causal link between tissue mechanics and transcriptional output, it would be beneficial to also show that ectopic tissue folding or stretch is sufficient to generate an increase transcription. Currently there is only data showing a loss-of-transcriptional upregulation when tissue folding is impaired, and it is possible that some of the mutants used to block gastrulation have broader pleiotropic effects that could impact Notch signaling.

We appreciate the concerns of the reviewer. However, because we have used a range of different methods to interfere with gastrulation, including mutations in the gastrulation specific Fog ligand, it is unlikely that these would all have the same pleiotropic effects. We have also tested a range of other mutations/disruptions including some that have quite pleiotropic effects but do not impact gastrulation, such as Zelda, and found that these do not alter the transcription profile. We have added information from some of the other genotypes tested (new Figure 2 – supplement 4).

We also agree that the converse experiment, testing the consequences of inducing ectopic folds, would be an excellent one. However, because we can only measure the read-out in the stripe of cells where Notch is active, this makes the experiment challenging- it’s difficult to know a relevant site for the ectopic folds and to induce them in the short timescale (aprox 10 min) needed to compare transcription levels before and after the fold and before gastrulation occurs. Nevertheless, we have made extensive efforts to carry out experiments along these lines over the past 2 years. Despite many attempts, we have not been able to induce ectopic invaginations, either earlier or in the relevant places, with the tools available in our experimental set-up. We have attempted to do this again during the review period but have not been able to do so successfully.

2. A discussion of the possible functional consequences of tissue movements on Notch-dependent transcription at this stage of embryogenesis would be helpful for the non-expert.

Consequences of perturbing the force-augmented transcription will likely be a less precise mesectoderm boundary, which has an important role in separating the flanking neurectoderm from either side as well as in forming the midline of the CNS. We have included a comment about the possible consequences from perturbing Notch target genes at this stage.

3. Data connecting the LINC complex to m5/m8 activity are rather weak, and I'm not sure these data in their current form add much to the paper.

The perturbation of the LINC complex has a robust effect on *m5/m8* transcription. This result was very striking because we had tested other possible intermediates, such as the Lamin related Kugelkern and the pioneer factor, Zelda, and had seen no change in transcription. We had omitted these negative data for simplicity, but these have now been added to the manuscript to give more context and illustrate that the result with LINC is noteworthy (new Figure 2 – supplement 4).

We have also included a description of the change in nuclear morphology and an analysis of the chromatin mobility during mesoderm invagination, in relation to gastrulation and the transcriptional changes (new Figure 6). The results show that there are substantial changes in nuclear morphology and that the chromatin becomes more mobile, a feature that is thought to be indicative of increased accessibility.

Reviewer #2:In the study presented in this manuscript, Falo-Sanjuan and Bray explore the connection between mechanical forces and transcription during gastrulation, a process involving extensive cell movement, rearrangement as well as tension sensing and specification of different cell lineages. The overall hypothesis and ideas are interesting and valid, the experiments are well designed and data are carefully analyzed. The study starts by explaining how the tension between cells can affect Notch signaling in the mesoectoderm through monitoring MS2 transcriptional activity from the m5/m8 enhancer in control and mutant embryos in which mesoderm invagination fails due to defects in cell-cell adhesion (α-catenin) or actomyosin regulation (RhoGEF2, concertina and fog). These results support the view that cortical cytoskeletal tension is causing an increase in Notch-dependent transcription. Then, somewhat abruptly the working model is revised when full-length Notch is shown not to be required for this increase (eve2-NICD in Notch RNAi); they hypothesize that if Notch signaling does not respond to tension at the cell membrane through ligand-receptor binding, the tension-sensing/gene upregulation at mesoderm invagination must relate to LINC complex function at the nuclear membrane. The transition to study of LINC complex function seems to be a bit jarring and needs more supportive evidence to better explain why alternative explanations were disregarded. Lastly, the title may be a bit misleading as the data support the view that "cell movements associated with gastrulation have a direct impact on transcription in the nucleus"… and that Notch-responses are not the only ones that are affected.

We are glad that the reviewer considers the hypothesis and ideas interesting and the experiments well designed. We have addressed the queries about the alternate models below and can add substantial data to address these.

The reviewer had specific concerns (detailed below) that relate to (i) quantitative analyses; (ii) there is an apparent contradiction between title and the fact that Notch appears dispensable for the observed phenotype; (iii) klar RNAi gives the opposite phenotype as expected (but somehow still stated as supporting their model) – this merits more explanation or investigation; and (iv) lack of attention is given to other explanations including possible connection between NICD-Twist-mechanotransduction.This is a very interesting study but the title, flow of the main text and its figures are somewhat misleading/hard to follow. In particular, the transition to study of LINC complex function seems to be a bit jarring and needs more supportive evidence to better explain why alternative explanations are disregarded. Also, the role of LINC complex, specifically klar in regulating tension-dependent transcriptional activity at gastrulation is unclear, more experiments need to be done to show how it interacts with NICD.

We have addressed the concerns below.

1) MS2 spot calling and signal quantification. The reviewer realizes that the MS2 quantification has been published before but for the current focus on levels of expression, the spot analysis should be revisited. Instead of identifying the MS2 spot, the authors appear to be using the maximum intensity within a nucleus. This ignores the shape of a spot and assumes that the maximum intensity sufficiently represents the spot. A better method of spot detection would include either gaussian fitting or thresholding. Since a key point of the paper is to study an increase in signal that is correlated to gastrulation, additional attention to quantification is necessary.

We appreciate the reviewer’s concerns. However, our analysis indicates that, at this stage in embryogenesis, the method used is very applicable way to capture the spot because there is little background or extraneous spots. To verify this, we have run the alternate analysis based on differential-of-gaussian filtering that integrates the fluorescence from the whole spot (Garcia et. al 2013) on several movies and obtained the same outcome, as illustrated in Author response image 1 for the cta and RhoGEF knockdown experiments. Because of the time needed to analyze all of the genetic conditions with the alternate approach (which is very computationally heavy), we have opted to use this simpler method, which we are confident gives a good proxy based on all the tests we have done.

**Author response image 1. sa2fig1:** 

2) Along these lines, the authors need to provide more information about the number of cells scored for graphs that show "fraction of nuclei…".

This information was provided in another panel but has now been added more explicitly in the relevant panels of each figure.

3) In Figure 2C,E, only embryos that show a clear change in mean levels of transcription are displayed. What is the total number of embryos collected for each condition? How was a "clear change in mean levels" defined?

This information has now been added to all plots that show correlations. We note that in most cases all embryos collected in each condition have been included (total n number is shown in the mean transcription plots). We had explained how a clear change in mean levels is defined in the methods, but we appreciate it may have been quite brief. We have now expanded on the definition and included a figure for clarity.

4) Figure 2, the control profiles of m5/m8 transcription activity in B, D, F appear to be very different, especially for fog mutants, is there a specific reason?

There is some variability caused by the genetic background, especially those that have a Gal4 present. Author response image 2 is a plot illustrating that, although there is a small amount of variability in the early phase, they subsequently all overlap throughout the period of MSE invagination, (45-60 mins), which is our focus here.

5) The authors used a vnd enhancer to show that mechanical forces have a general effect on transcription at gastrulation. However, there is no further attempt to explore or discuss how vndEEE might be regulated, through another tension-sensing transcription factor or maybe also by changes in nuclear property/LINC complex? Twist seems likely (see pt 8 below).

Our hypothesis is that the same mechanisms are involved for *vndEEE*, as this would be the most parsimonious explanation. We agree that it could be valuable to test *vndEEE* in other mutants, but have not done so, we focussed on demonstrating that the mechanism is not via Notch and hence points to a more general mechanism. We agree also that Twist was a prime candidate, but as discussed below, our evidence does not support that.

6) Figure S4, if m5/m8 is Notch-dependent, what causes the increase at time 0 in the mean profile of Notch RNAi?

We have previously shown that *m5/m8* expression is compromised when Notch activity is perturbed by a range of different mutations (Falo Sanjuan et al., 2019). Notably, in *neuralized* mutants there is no expression of *m5/m8* remaining. In order to carry out the experiments here, we used NotchRNAi to perturb Notch activity, as it was the most genetically tractable for the experiments (e.g. *neuralized* mutants could not have been genetically combined with the MS2 system and ectopic NICD production). Unfortunately, RNAi is not always fully penetrant and some embryos retained residual activity. As evident from the plots in Figure 3 – supplement 1B, expression is completely or almost completely abolished in 6/8 embryos. We have opted to include all of the data for transparency, but include now also a plot from those embryos where we are confident the knock-down is effective.

7) It is interesting that the ectopic expression of eve2-NICD in Notch RNAi background rescues m5/m8 transcription activity within the eve stripe (Figure 4), but how much of this increase relates NICD versus other tension activated factors (e.g., twist)? Is twist upregulated in the domain of eve2-NICD expression in Notch RNAi embryos? How does this study/timing of m5/m8 upregulation at invagination relate to the mechanical signals discussed in Pouille et al., Sci Signal 2009 (which should be cited).

We have previously shown that *m5/m8* transcription is dependent on Notch and that Twist is involved in priming so that the enhancer response in a sustained and robust way. Like the reviewer, we wondered whether Twist might be involved in directing the enhanced expression at gastrulation, in part based on the report that Twist is sensitive to mechanical force in other parts of the embryo, in other work from the Farge group. We have invested a large amount of time to follow up on this model but can find no evidence that Twist is at the nexus. For example, we carefully quantified Twist levels throughout nuclear cycle 14, and there is no increase at the time of mesoderm invagination. These data have now been added to the manuscript as new Figure 6. Furthermore, a mutant version of the *m5/m8* enhancer that lacks Twist motifs is still upregulated at the time of mesoderm invagination. This point has been made more explicit in the text (see page 18).

The mechanical signal described in Pouille et al. would be upstream of the events we are investigating as it altered endocytosis of Fog, the ligand that initiates mesoderm invagination. We have made this point more explicit and included the reference as requested (see Page 16).

8) The conclusion that LINC complex/Klar plays a role in tension-regulated Notch signaling activity is poorly supported. Figure 5B shows that there is an early increase in m5/m8 activity, which is, as the authors point out, opposite from the expected result. Therefore, it is necessary to better define the relationship between NICD and nuclear properties modified/influenced by LINC. How does loss of function klar (or other LINC protein) impact intracellular Notch signaling and other tension-related processes, including localization of α-catenin at the AJs, RhoGEF2 at the apical membrane, actomyosin pulsing, expression of cta and fog, etc. Depending on whether twist is found to be upregulated by eve2-NICD (see point 7 above), it might also be worth looking to see how twist expression (upregulation by NICD?) is affected by klar.

We argue that LINC is required to transmit the consequences from tension regulated forces, generated by Fog and actinomyosin changes, onto the nucleus. There is no evidence that Klar would alter any of these upstream steps in the mechanotransduction. In contrast there is growing evidence that LINC is important in transducing forces to the chromatin. We have thus focussed on potential downstream changes in the nuclear properties.

These additional results are reported in a new section “Changes in nuclear properties at gastrulation”. First, we have measured nuclear entry of different transcription factors, including Twist. We not find any changes in nuclear levels that can provide an explanation as shown by the new data that we have added (new Figures 6, Figure 6 – supplement 1). Second, we have quantified additional features such as nuclear morphology and chromatin motility. Both these properties change at gastrulation and could contribute to the effects seen (new Figures 6, Figure 6 – supplement 2 and Figure 6 – supplement 3).

9) Unlike m5/m8 enhancer which requires Notch signaling input directly to help activate gene expression, Notch input at the sim enhancer acts to 'derepress' expression by eliminating Su(H) repressor input and allowing output to be responsive to Dorsal and Twist (Zinzen et al., Dev Cell 2006; Morel & Schweisguth, Genes Dev 2000). Therefore, the roles of Notch in m5/m8 and sim enhancers differ. In the current study, the focus is m5/m8 with little done to analyze sim enhancer properties. Interestingly, in Morel & Schweisguth, when Su(H) sites are mutated in a sim enhancer (sim.mut), reporter expression remains in N mutant embryos. It is likely that the sim.mut enhancer would also show upregulated gene expression at mesoderm invagination if tested; and if so may be a better test case for tracking the input that relates to mechanotransduction with subsequent mutagenesis. Possibly this input is Twist.

Our evidence suggests that NICD also has a positive effect on *sim* enhancer, as well as alleviating repression. We note that, although there is some expression remaining in the absence of Notch, this is very variable and weak in comparison to the wild-type. In addition, the *sim* enhancer shows a very similar upregulation to *m5/m8* at the time of mesoderm invagination, which is lost in embryos where α-Catenin is depleted. We have added these additional data with the *sim* enhancer to the manuscript (new Figure 2 – supplement 3). As discussed above, we do not find evidence to support the hypothesis that Twist is the primary transducer.

10) Farge 2003 already showed that Twist increases residency in the nucleus downstream of mechanical induction. Relating to comments above (e.g. point 7 and 9), it is likely that the NICD-effect could be indirect, working to upregulate Twist. If so, this study is still interesting, but are the current results novel?

As discussed above, we also thought Twist would be a good candidate based on Farge 2003 and have investigated this possibility quite extensively. We do not find any evidence for a change in the levels of nuclear Twist at this stage, having used a nano-body to measure the levels throughout nuclear cycle 14 (see above and new Figure 6). Furthermore, an *m5/m8* enhancer in which all of the Twist motifs have been mutated (which exhibits much lower levels of expression) still undergoes a marked increase in activity at the time of gastrulation (Falo-Sanjuan et al. 2019). This point has been emphasised more clearly in the text (page 18).

11) The title is confusing as the data support the view that "cell movements associated with gastrulation have a direct impact on transcription in the nucleus"… and that Notch-responses are not the only ones that are affected.

We have modified the title to take on board the concerns and we think this better describes the results obtained.

Reviewer #3:This manuscript is of broad interest to readers in the fields of Cell and Developmental Biology. The authors provide an elegant set of experimental evidence demonstrating that Notch transcriptional activity is modulated during *Drosophila* mesoderm movements. The data are of very good quality. The authors could boost their conclusions by better replacing them in the context of their previous studies regarding the role of adherens junctions in Notch transcriptional regulation or turnover (Falo-Sanjuan and Bray, Development 2021). The findings on Klar mutant conditions are highly relevant but more difficult to unit with the interplay between Notch transcriptional control and mesoderm invagination.The major strength of the manuscript is the quantitative and original analyses of Notch (N) transcriptional levels in conjunction with morphogenetic movements in multiple genetic backgrounds. The authors could improve their manuscript by addressing the following points:1. In Figure S1, the authors have also quantified N activity with two other reporters of N transcriptional activity. Both reporters show an increase at 50 mins as described in the main text, but both reporters also show a decrease at 55 mins. This decrease is not observed for the line (II) used through the manuscript. One could therefore argue that this drop is also a relevant characteristic of Notch activation during mesoderm invagination. This data should therefore be placed in the main figure and the decrease should be mentioned in the main text. In addition, it will be important that the authors performed the some loss of function analyses (for example RhoGEF2, Fog and Notch mutant conditions) using these two other reporters to further validate their findings.

The decrease noted by the referee occurs after gastrulation is completed when the midline stripes come together and the cells in the mesectoderm divide. Our evidence suggests that Notch signalling ceases around that time and that the time it takes to complete this is more variable across embryos from the same genetic condition. It is also the time when there are extensive movements and sometimes the tracking is lost. This leads to considerable variability between movies, especially in genetic conditions where gastrulation is a bit delayed or stuck. We can confirm that both the other reporters have been examined in Notch loss of function conditions, where their expression is lost. We have also tested their behaviour in embryos where α-Catenin is depleted and both show similar loss of upregulation to the main reporter used in the paper. We have added these data to the manuscript (see new Figure 2 – supplement 3).

2. Page 7: The following statement is confusing. "We note the mean levels of transcription from this reporter were not significantly reduced prior to gastrulation, unlike those from another reporter insertion and the endogenous genes that showed decreased levels also during cellularization (Falo-Sanjuan and Bray 2021). The difference likely arises because this m5/m8 [II] insertion achieves maximal transcription at lower levels of Notch activity." In the cited previous paper, the authors have shown that a-cat affects Notch transcription during cellularization. It is unclear why a difference in reporter strengths will lead to distinct conclusions. This needs to be further experimentally explored and explained.

In all the experiments we have done, the second chromosome *m5/m8* reporter gives a more robust response. We too have been perplexed by the reasons for this, although it is well known that different genomic insertion sites of the same reporter frequently yield slightly different responses. We have recently generated a new CRISPR line where the endogenous gene has been tagged with MS2 loops and verified that it shows the same transcription profile. We also find that the other enhancers yield similar results after gastrulation, making us confident that the effects are not enhancer specific, albeit sometimes the magnitude of the effect may vary. We have explained this more clearly in the text (see pages 7-8) and have added the additional data for the other enhancers (new Figure 2 – supplement 3).

3. Page 12: "As predicted, this treatment greatly reduced transcription from m5/m8 [II] (Figure S4AB).". The reduction is 2-fold. How these fold changes compare with the ones observed in RhoGEF2 and Fog mutant conditions? The Notch mutant conditions should be shown in figure 1. Furthermore, one could argue that a large part of the increase in reporter expression does not reflect an increase in Notch activity. Please discuss.

Here we are a victim of the variability from RNAi knock-down experiments. We have previously used several different mutations to compromise Notch activity, including *neuralized*, where there was no residual expression from the reporters (as previously published, Falo Sanjuan et al., 2019). Here we used NotchRNAi to perturb Notch activity, as it was the most genetically tractable for the experiments. Unfortunately, RNAi is not always fully penetrant and some embryos retained residual activity. As evident from the plots in Figure 3 – supplement 1B, expression is completely or almost completely abolished in 6/8 embryos. We have opted to include all of the data for transparency, which is why some residual expression is seen in the mean plots, as noted by the referee. We have now included a plot from those embryos where we are confident the knock-down is effective to illustrate that very little transcription remains (Figure 3 – supplement 1A, new panel) The fold changes are very different from other knock down experiments, where there is transcription remaining in a significant proportion of the cells in all the embryos, as is evident in the heatmaps. We hope this clarifies the point.

4. Figure 4. It will be important that the authors plot on the same plot with the same units the endogenous Notch activity in both wt and NotchRNAi.

We are unclear what the reviewer is requesting here, and what the concerns are. In each experiment we compare with a wild type control from embryos imaged under the same conditions. For all of the Notch RNAi experiments there is a plot of the equivalent control, with control RNAi, from the same experiments. If comparing between very different experiments (e.g WT and Notch ICD overexpression), the time and imaging conditions may not be identical. For example, some experiments were conducted pre-Covid closure and others after, with changes in the microscopes between. A laser burned out at one point and some experiments were performed on a different microscope. However, where these changes have occurred, controls were always imaged in parallel. This is why there are sometimes slight differences between the control plots.

5. As stated by the authors, Notch transcription findings in Klar mutant conditions are distinct from the authors' prediction since transcription increases prior to mesoderm invagination. The role of nucleus deformation and the Klar contribution remain difficult to understand and insert in the authors proposed model.

We agree that the results are not 100% straightforward. However the data argue (1) that integrity of LINC complex is important for the normal levels of gene expression and (2) that the levels of transcription are no longer sensitive to the forces from gastrulation. As we have tested other factors and not found any other that prevents the gastrulation related increase, without compromising transcription overall, we therefore consider this to be a substantial result.

6. Statistics: Could the authors propose statistical tests to further support their findings regarding temporal changes in reporter levels and differences between distinct mutant backgrounds?

We have done this for the correlation plots and provided the results in the R-value. The mean plots are significantly different from one another, as is evident from the error bars. As discussed we always include all the data in the plots, to show the variability within a given genotype. This has been the standard approach in the current literature.

[Editors’ note: what follows is the authors’ response to the second round of review.]

Essential revisions:Overall, the manuscript is much improved and the authors have addressed most of the reviewers' major concerns. There are two remaining issues that require clarification and additional technical information related to imaging conditions and corrections for nuclear depth over the course of gastrulation, and a previously reported increase in transparency of klar embryos.1) Please provide additional information about the imaging pipeline – specifically whether any corrections are being made for spatially-dependent changes in signal due to movement of cells in the z-direction. See reviewer 2 comments for details.

We appreciate the comments of the reviewer, because the potential for the movement of the nuclei to be a confounding factor was one that concerned us at the outset. All the movies were carefully evaluated to ascertain whether there was any relationship between the nuclear positions and the change in MS2/MCP signal intensity. We found no correlation. We have now added the graph showing this lack of correlation to the supplementary data (new figure 1S1E ). We note, as discussed in more detail below, that the strategy proposed by the reviewer, to normalize using the histones, is not feasible due to variable changes in the levels and the chromatin density etc, that complicate any such quantifications. We hope that the additional information provided addresses the concerns.

2) The klar mutant background is associated with a clearing of embryos, at least in older embryos at stage 8 (Supatto et al. Nature Protocols 2009). Please provide information regarding whether there is an increase in transparency of klar embryos at NC14 that may result in the GFP signal appearing brighter that wild type controls.

We are grateful to the reviewer for highlighting this possibility. Indeed we have now checked and there is an increase in GFP detection in the Klar mutant embryos. We have carefully quantified this difference, and corrected for it. However, the effect is a small one with a very minor impact on the overall levels that does not alter the results (it makes a small adjustment in our favour).

Reviewer #2:The manuscript is improved. In particular, the data are more clearly presented and the authors have added more experiments, namely testing whether levels of Twist, chromatin import, chromatin compaction and/or accessibility relates to the increase in mesectoderm gene expression identified at the time of mesoderm invagination.

We are glad the reviewer appreciates the additional experiments testing the role of Twist.

The author's testing of Twist addresses one of the reviewer's major concerns. While it is still possible that post translational modification of Twist acts to increase its activity at invagination, their dismissal of Twist for the purpose of this study is reasonable based on the data presented.Another major concern was that the use of maximum intensity may lead to noise in the spot calling numbers. The authors state that using the maximum intensity is an accurate method because of tests they have done, however they do not elaborate on what these tests are or how they have verified this beyond the limited use of the alternate method from Garcia 2013. While using the method in Garcia 2013 does show the same general trend, the error is much larger using this method and this doesn't completely support the claim that this method proves the simpler method is correct. The fundamental concern is that single, bright pixels that occur outside of the actual MS2 spot could be skewing the data. An easy way of dealing with this is to apply a median filter, which will remove "salt and pepper" noise to the images. The authors note that they apply this filter across time to smooth the data, however this is would be selecting for nuclei that tend to be on continuously. Another approach would be to manually confirm that the maximum pixel occurs within a spot, but this would be tedious. In the methods section it is stated that "subsequent analysis allows to classify confidently when the maximum value is really representing a spot," however it is unclear how the subsequent analysis alleviates this concern. At minimum, this section of the methods needs to be expanded and made clearer so that the reader can evaluate the method being used.

We apologize if the details of the methodology were not sufficiently clear. As the reviewer notes, we have applied a median filter to remove noise from background pixels that are not coming from an MS2 spot (“salt and pepper”). The wording we used was perhaps misleading because the filter is only applied over 3 frames, i.e. over a period of 45-60”. Thus, it is not selecting for “continuously active spots”. The time-window was chosen because, based on transcription rates and minimum signal needed from MS2 loops, bursts shorter than 1 min would probably not be detectable so any bright pixels present for such a duration have a high probability to be background. Conversely, the “subsequent analysis” involved applying background subtraction, median filtering and selecting spots classified as ON for 5 frames (1.25-1.6 min) to identify active sites of transcription. These parameters were validated by overlaying nuclei defined by the pipeline as actively transcribing with the MS2 signal, and confirming that the two were concordant. These aspects of the analysis have been explained more clearly in the Methods (page 34).

I do still have a worry that their imaging pipeline is not properly correcting for spatially-dependent changes in signal due to movement of cells in the z-direction. Because they have a H2A-RFP associated with the MCP-GFP MS2 movies, they should use the RFP signal to background correct (not the MCP-GFP signal associated with inactive nuclei). The H2A-RFP signal can vary in space in time due to position of nuclei. The authors are imaging the MS2 signal in a tissue that moves, however they don't provide a correction for this movement or rule out whether this movement is changing intensity of the signal. Specifically, when signal is lower, the MS2 is in a more dorsal location than compared to after invagination where the signal is at the ventral most point. Did the authors correct for any difference in intensity that could be occurring from differences in the position of the tissue? It is conceivable that signal from the ventral most point will be more optically accessible than from points located more dorsally, due to the curvature of the embryo and possibly the diffraction of light as it moves through tissue to reach the objective. An easy way to correct for this would be to use the signal of the H2A signal that was used to segment and track the nuclei, as any positional effects should also affect the H2A signal, and the H2A signal is already mapped to its corresponding spot.

We acknowledge that tissue depth and position of the MS2 spot in Z might affect the intensity being detected, however we had carefully evaluated the data and had ruled out the possibility that the transition in levels is due to changes in nuclear position. We have added data to the supplementary (new figure 1S1E) to illustrate this point.

We have been able to analyze the relationship between MSE nuclear depth/curvature and the MS2/MCP signal because individual embryos are mounted with slightly different positions so that each has differing changes in Z. In most cases, the embryos are not mounted completely ventrally, which means that the MSE is very close to the coverslip before as well as after gastrulation. As the embryo gastrulates, it does so within the vitelline membrane so that the MSE is captured in a similar Z position throughout time. This means that, in these cases, the signal detection is not very affected by curvature of the embryo and the depth of imaging. In other cases where the embryo is mounted more ventrally the difference in Z positions is only 5μm on average. By comparing the data from each embryo, there is no correlation between how far the position of the MS2 spots differed in Z before and after gastrulation and how much levels increased. We initially made this analysis for all the original movies (used for Figure 1G) and have now performed the same analysis for the new movies we generated for the gastrulation studies (Author response image 3) . In neither case is there any correlation between the change in Z position and the change in levels. We have also explored these parameters carefully on an embryo by embryo basis and are confident there is no confounding effect from the nuclear depth. We have now added the left hand graph to supplementary Figure (Figure 1S1E) and commented on this point in the text (page 6).

**Author response image 3. sa2fig3:** 

We also note that in other genetic conditions, such as RhoGEF2 depletion, where tissues move from the same positions in DV and Z as wild type embryos, (only slower), we do not detect any increase in levels. If the increase was due to differences in Z positions, a similar increase would still be manifest in these embryos, which it is not.We appreciate the suggestion from the reviewer to utilize the His2Av-RFP intensity as a way to normalize from any changes in depth. However, in our experience this would not be a good method for several reasons. First, its levels vary over time, independently of Z position, because as nuclei increase volume the His2Av signal becomes dimmer. Second, heterochromatin begins forming during nc14, giving rise to a much more heterogeneous signal that includes differences along apical/basal polarity. Thus while His2Av is very valuable for segmenting the nuclei, its variability during nc14 makes it unsuitable for correcting on a nucleus-by-nucleus basis for the subtle changes in intensity due to changes in Z position.

In addition, the klar mutant background is associated with a clearing of embryos such that the GFP signal may appear brighter because embryos are more transparent; this has been shown for older embryos at stage 8 (Supatto et al. Nature Protocols 2009) but it is unclear if this is the case at nuclear cycle 14. If klar mutant embryos at these earlier stages are more transparent, could this account for the apparent increase in GFP signal associated with the klar mutant background (Figure 5B)? Might the H2A-RFP signal serve to normalize the ctr RNAi vs klar RNAi data? Is the H2A-RFP signal also increasing in the klar RNAi? If that is the case, and the klar data is more similar to wildtype – the early phase of transcription may need to be offset vertically (allowing it to align with wildtype). In this alternative scenario, it would appear then that there is a transition late in the klar mutant background. Might the clearing of lipids affect timing of mesoderm invagination and/or when it is recognized/called?

We thank the reviewer for bringing this reference to our attention. *Klar RNAi* embryos indeed exhibited higher GFP and RFP signals. We quantified this difference by measuring the average His2Av-RFP signal (15.8 % lower in control embryos) and average MCP-GFP signal in nuclei that never show MS2 activity (11% lower in control embryos). We have as a consequence corrected the MS2 signal for the “clearing effect” by reducing the signal in *Klar RNAi* by 11%. This correction has a very modest impact which doesn’t change the overall result. These data are now included in supplementary Figure 5S2 and commented on page 17.

The reviewer appreciates the premise of this study but the authors would be wise to do a bit more controls/investigating to ensure that the observed upregulation in signal associated with mesoderm invagination is not an imaging artifact: (i) might positional effects on MS2 signal be responsible for the increase in GFP observed when cells invaginate/are more directly illuminated; and (ii) to determine whether klar mutants are more transparent and if so use methods to normalize signals obtained in klar in comparison with wildtype.

We have addressed these points in detail in the two sections above.

Reviewer #3:The authors have improved their manuscript and addressed my points. They have added a lot of data related to the dynamics of the nucleus and chromatin mobility. Most of these results are negative but they contribute to the detailed characterization of the process. Overall, the findings support the authors' conclusions.

We are glad that the reviewer appreciates the additional data and that these have addressed many of the points they raised.

Nevertheless, the role of Klar remains difficult to fit in the authors' proposed model, since Klar loss of function phenotype is distinct from the ones observed in mutants affecting the mechanical forces associated with gastrulation (α-cat, RhoGEF2 and Fog). I therefore doubt that Klar would be part of the mechanisms by which gastrulation associated forces modulate Notch-dependent and -independent transcriptional regulation.

We appreciate the reservations of the reviewer, but we disagree with the premise that a difference in phenotypes a priori rules out a role in the mechanism. The *Klar* mutants share a key element of the phenotype from α -Cat, RhoGEF2 which is to depress the increase in transcription. Klar exhibits additional changes which would suggest it has other roles besides transducing the mechanical signal.